# MuSED-FM: A Benchmark for Evaluating Multivariate Time Series Foundation Models

## Abstract

Multivariate Time Series Foundation Models (TSFMs) aim to identify patterns in multiple contexts to make meaningful predictions of the future. At their core is multivariate capability; models make use of information from multiple sources rather than relying on a single signal with limited information. Learning multivariate models requires meaningful evaluations, but current benchmarks are limited in two key ways: quantity and quality. There are a limited number of multivariate time series datasets, with existing ones lacking size and diversity across domains. Furthermore, although some collections of time series might be marketed as multivariate, it is not proven that they contain meaningful information in multiple contexts. This work takes a major step in both directions, providing a Multivariate Time Series Evaluation Dataset for Foundation Models (MuSED-FM). MuSED-FM spans 16 multivariate time series domains and introduces novel synthetic data techniques, comprising 67 billion data points and 2.6 million time series. To improve and prove the quality of multivariate data, we provide a powerful suite of benchmarking tools focused on identifying the multivariate predictability of a time series and introduce novel multivariate predictability aggregate metrics based on classical methods. Finally, we evaluate current state-of-the-art TSFMs for both univariate and multivariate capability, finding that despite multivariate predictability identifying correlation, univariate prediction often matches or outperforms multivariate prediction across models.

## 1 Introduction

Accurate time series forecasting is valuable in a wide variety of domains, such as scientific advancement [96], finance [88], operations analysis [89], and weather [55]. Catalyzed by recent advances in foundation models for vision and language [38; 63], Time Series Foundation Models (TSFMs) [30] achieve state-of-the-art forecasting accuracy by identifying patterns in-context. However, the history of a *single* time series, that is, a single sequence of numbers, often lacks sufficient information to forecast all but the simplest trends. Incorporating information about the patterns in other, covarying time series' is essential to improve forecasting accuracy. Thus, it is vital that TSFMs take advantage of multivariate data.

Due to the necessity of multivariate TSFMs, it is imperative that time series forecasting evaluations reflect multivariate forecasting performance. However, evaluating the multivariate capabilities of TSFMs remains a challenging problem. Existing open multivariate time series datasets suffer from two key issues: quantity and quality. First, the quantity and diversity of data is insufficient. Existing TSFMs are trained on limited domains of multivariate data, leading to poor generalization to new multivariate domains. Second, the quality of existing multivariate evals is poor. Although existing multivariate datasets provide multiple contexts for time series forecasting, they fail to demonstrate that the additional contexts contain valuable information for forecasting.

Motivated by the challenges of multivariate time series evaluation, we introduce a **Mu**ltivariate Time **S**eries **E**valuation Dataset (MuSED-FM) consisting of a significant number of multivariate datasets from a wide range of sources. We aggregate data first from open source multivariate time series benchmarks on a wide range of domains and frequencies, from climate information to web data, scientific domains, or government statistics. We then take univariate data from similar domains, such as finance or weather, and construct multivariate time series data by combining series with similar

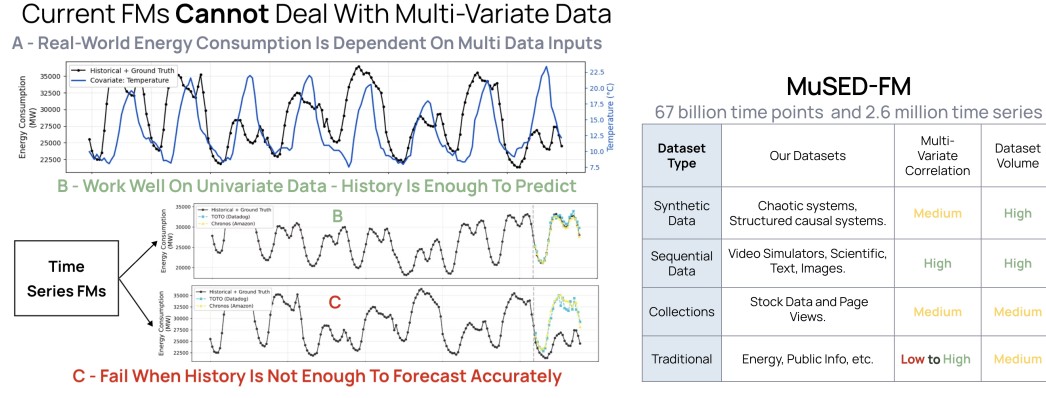

(a) Current TSFM Limitations        (b) Dataset Categories

Figure 1: **Limitations of TSFMs and our proposed MuSED-FM Left:** Real data highlight a core limitation of current TSFMs: they perform well when the past alone suffices (B) but fail when prediction depends on exogenous, multivariate signals (C). In (A), energy demand depends on temperature; in (B), Chronos [5] (yellow) and Toto [29] (blue) succeed on a univariate case; in (C), the same models largely ignore covariates and default to seasonal extrapolation. **Right:** MuSED-FM is a curated multivariate benchmark—spanning sequential, synthetic, real-world, and combined datasets—designed to rigorously evaluate TSFMs' ability to use cross-variable information.

descriptions. Then, we augment this data by representing various sequential data, including video, controls, and robotics data, which can be transformed into a usable range for multivariate foundation models. Finally, we include synthetic data generated according to either a distribution of multivariate ordinary differential equations [68], or using a structural causal model prior [56]. In Figure 1, we provide a categorization of the kinds of information present in the dataset and the predictive properties of the datasets by domain. In particular, this evaluation benchmark can answer: 1) How well does a multivariate TSFM perform on a wide range of realistic time series data? 2) How does a model's multivariate performance compare with univariate performance? 3) How can one assess whether a novel multivariate dataset actually exhibits univariate *and* multivariate predictability? This work provides comprehensive foundation model evaluation on several existing foundation models to answer the first two questions, and a comprehensive univariate and multivariate predictability analysis to answer the last. The evaluation and analysis in this work illuminates the core premise: *Current multivariate time series foundation models ubiquitously fail to utilize multivariate information, and MuSED-FM is the key evaluation necessary for multivariate capability.*

In aggregate, the contributions of this paper are as follows:

1. **We propose MuSED-FM, the largest collection of multivariate time series datasets**, including open-source, univariate datasets with correlated information, multivariate series derived from multimodal data (including scientific, simulator, video, vision, and language data), and multivariate synthetic time series generated via multiple novel techniques. This dataset includes 67 billion time points, 2.6 million series, and an average of 26 variates across 45 datasets and 16 domains.
2. **We provide a novel metric** reflecting the univariate and multivariate "predictability" of a time-series, utilizing classic multivariate relational tools: transfer entropy [109], Granger Causality [52], Convergent Cross Mapping [115], lagged cross correlation [10] and DLinear [142].
3. **We evaluate current state of the art TSFMs**, illustrating that current state of the art multivariate models fail to exceed univariate performance across multivariate data (Figure 2).

## 2 RELATED WORK

TSFMs have emerged recently, with pioneering efforts [103; 44; 105] applying principles from language [38] and vision [63] to the field of time series, sometimes explicitly [53]. Previous datasets [84; 85], [7], [31], [47], served primarily to access domain-specific training data rather than as an evaluation of a single model operating across multiple domains. However, the subsequent capacity of foundation models [73; 91; 127] as tools that offer versatility and even performance over classical methods [90; 140] such as ARIMA or VARMAX, has inspired recent investigation into time

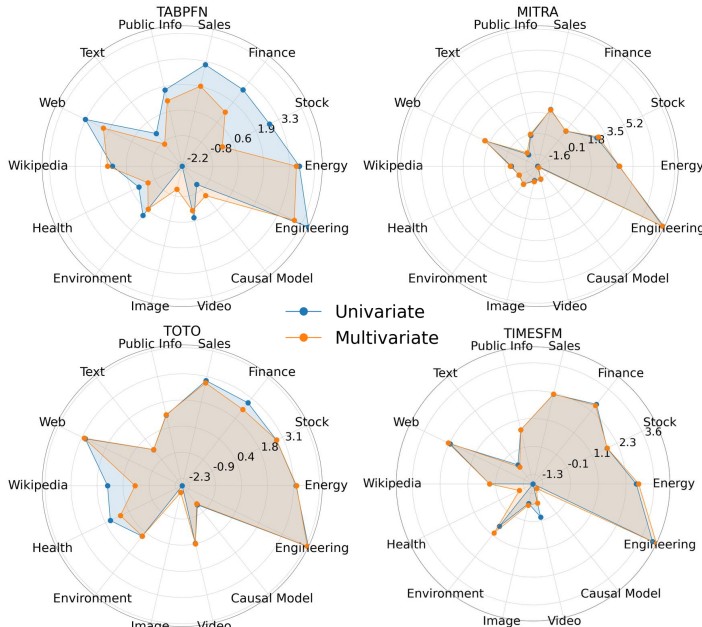

Figure 2: **Existing State-Of-The-Art (SOTA) Multivariate TSFMs fail to effectively use the available correlated information.** We showcase the comparison of the univariate and multivariate performance of 4 SOTA models on 14 diverse domains. Specifically, we plot the logarithm of the inverse of the Mean Average Percentage Error (MAPE) metric (higher is better). Surprisingly, providing additional correlated information (multivariate case) leads to similar or worsening of the model performance (TabPFN), with marginal improvements observed in very few cases.

series foundation models [133; 42; 139; 131; 43; 5; 36; 50; 74; 51; 113]. Nonetheless, initial efforts [105; 5] focused on univariate time series', which are both simpler to process and more readily available. However, univariate time series', by definition, do not encode contextual information, which limits their predictive power. Thus, recent work has begun to consolidate on the question of Multivariate Time Series Forecasting [139; 16; 29; 6; 105; 57], though in this work we demonstrate that these results, while promising, suggest that multivariate forecasting remains an open question. Works in foundation modeling often utilize large, privately owned datasets to train models, such as [29; 81; 105]. This work contributes to the effort for a multivariate time series foundation model by **1)** providing the largest (number of data points, rows and correlates) and most diverse (spanning multiple domains and classes of data) multivariate time series evaluation benchmark and **2)** providing aggregated metrics and benchmarking for *identifying multivariate predictability* of both models and data.

Meaningful evaluation and training data is the cornerstone for a multivariate time series foundation model, and while main benchmarks remain domain specific, focusing on classic fields such as energy [151; 128], weather and climate [134; 55], traffic [17; 79; 92], finance [88; 58], air quality [147], healthcare and brain signals [32; 62; 126; 96; 136] but also branching out into compute or network related domains [117; 130; 87; 75], cryptocurrency [4], biology [97], industrial devices [125], icicle formation [19], aerospace [144], action identification [49], environment [54]. Alternatively, many benchmarks focus on related problems such as anomaly detection [94; 59; 78; 135; 101], classification [98], robustness [146], etc. The substantial diversity of domains offers both a tantalizing promise and an obvious challenge for time series foundation models: the promise of widespread applicability and the challenge of myriad diversity. Nonetheless, recent work foundation model benchmarkes include [84; 119; 85], but subsequently with [3; 47; 9; 1; 71], as well as benchmarks focused on balanced or fair assessment [99; 112], horizon [145], multimodality [76], irregular sampling [15], or dynamical systems [68; 120]. Other work focuses on benchmarking with different foundation models, including [70; 111; 77; 100], while also contributing some compilations of data. While these methods often include multivariate series, they often do not focus on the multivariate problem. First, this can lead to a limited volume of multivariate data, and datasets such as those used in [47], [1], and [70] are either entirely or heavily univariate. Furthermore, the analysis of the mul-

| | Domain | Counts | Number of Timesteps | Average Number of variates | Dataset Count | Number of Time Series |
|---|---|---|---|---|---|---|
| **Traditional** | Energy | 114,941,077 | 4,801,760 | 19.32 | 34 | 315 |
| | Public Info | 4,625,412 | 219,946 | 17.00 | 10 | 270 |
| | Health | 202,815 | 41,756 | 8.00 | 2 | 9 |
| | Sales | 280,348 | 29,275 | 16.83 | 6 | 15 |
| | Climate & Env. | 496,270,478 | 10,993,235 | 24.86 | 7 | 54 |
| | Engineering | 1,975,188 | 183,200 | 8.50 | 2 | 20 |
| | Finance | 99,666 | 6,328 | 15.75 | 8 | 8 |
| | Web | 15,169 | 2,167 | 7.00 | 1 | 1 |
| **Collections** | Stock | 8,852,800 | 354,112 | 25.00 | 1 | 352 |
| | Wikipedia | 18,084,600 | 602,820 | 30 | 1 | 1,530 |
| **Sequential** | Images | 126,720,000 | 11,520,000 | 11.00 | 1 | 20,000 |
| | Text | 190,308,864 | 10,016,256 | 19.00 | 1 | 19,563 |
| | Video | 9,403,332 | 154 | 10 | 1 | 6114 |
| | Scientific | 1,373,811,912 | 26,441,444 | 54.00 | 5 | 48,812 |
| **Synthetic** | Dynamic | 5,142,085,632 | 214,253,568 | 25 | 1 | 209,232 |
| | Causal Model | 57,404,227,584 | 1,808,007,168 | 25.4 | 5 | 2,354,176 |
| **Combined** | | 64,791,804,797 | 2,077,472,139 | 25.7 | 85 | 2,680,671 |

Table 1: **High-Level Dataset Summary by Domain.** We provide a breakdown of our benchmark, **MuSED-FM**, including the average number of variates, time series, and datasets in each domain.

tivariate datasets in [1] is limited to mostly a small subset. Of existing time series foundation model datasets, [71] shows the closest resemblance to MuSED-FM, containing multiple real-world time series datasets. Our work subsumes this contemporary work by first including a greater volume of datasets spanning several more domains and time series patterns. Second, including novel datasets by combining related univariate series, transforming sequential data into time series and generating synthetic data to create a more comprehensive datasets, which we discuss further in Section 4. Third, the analysis of variates in [71] uses pearson correlation metrics, which only measure if two series look similar, regardless of how useful the variates are for prediction. By contrast this work introduces a novel multivariate *predictability* score discussed in Section 5. Overall, MuSED-FM exposes the limitations of existing multivariate models, provides meaningful correlation analysis tools and provides a much greater volume and scope of multivariate data. Extended discussion of related work can be found in Appendix A.

## 3 MULTIVARIATE TIME SERIES FOUNDATION MODELS

The multivariate time series forecasting problem involves the correlate variates time series $\mathbf{x} \in \mathbb{R}^{T \times d_x}$ and target variate $\mathbf{y} \in \mathbb{R}^{T \times d_y}$, where $d_x$ is the number of correlate variates, $d_y$ is the dimension of the target variate and $T$ is the number of timesteps. The time series is then separated into the context (history) component $0 < h < T$ and the prediction component $p = T - h$, where we represent $\mathbf{x}^{:h}, \mathbf{y}^{:h}$ as the context components of correlate and target variates respectively, and $\mathbf{y}^{-p:}$ as the predict component, and $X^{:h}, Y^{:h}, Y^{-p:}$ as the random variables representing the distribution of multivariate time series correspondingly. The objective of a multivariate time series forecaster is to learn a model of the conditional distribution: $f(\mathbf{x}^{:h}, \mathbf{y}^{:h}) \to \mathbb{P}(Y^{-p:}|X^{:h} = \mathbf{x}^{:h}, Y^{:h} = \mathbf{y}^{:h})$. Foundation models for time series train the model on the space of all multivariate series to $(\mathcal{X}, \mathcal{Y})$ to perform the best estimate of this model. By contrast, a univariate time series forecaster simply models $f_{\text{uni}}(\mathbf{y}^{:h}) \to \mathbb{P}(Y^{-p:}|XY^{:h} = \mathbf{y}^{:h})$, assuming that $d_y = 1$.

To properly situate the datasets in this work, we provide univariate analysis, multivariate analysis and benchmarking with existing foundation models. Univariate analysis involves tools such as Entropy, Fourier analysis, etc., as described in Section 5, and aim to characterize the distribution of $\mathcal{Y}$, not only for particular domains, but also for synthetic data. By contrast multivariate analysis aims to assess the amount of relative information in the history of the variates $X^{:h}$ with the future of the target $Y^{-p:}$, which can be formalized as the time-based conditional mutual information $I(X^{:h}; Y^{-p:}|Y^{:h})$ [110]. While the distributions of all three random variables are unknown in practice, our analysis offers a suite of approximation

techniques for estimating this quantity. Finally, benchmarking assess the performance of the models $\mathbb{E}_{\mathbf{x}^{:h},\mathbf{y}^{:h},\mathbf{y}^{-p:}\sim\mathcal{D}_{\text{eval}}}\left[\left\|\mu(f_{\text{model}}(\mathbf{x}^{:h},\mathbf{y}^{:h})) - \mathbf{y}^{-p:}\right]\right\|\right]$, where $\mathcal{D}_{\text{eval}}$ is the evaluation dataset stratified by domain. We also compare the performance of univariate and multivariate forecasting $\mathbb{E}_{\mathbf{x}^{:h},\mathbf{y}^{:h},\mathbf{y}^{-p:}\sim\mathcal{D}_{\text{eval}}}\left[\left\|\mu(f_{\text{model}}(\mathbf{x}^{:h},\mathbf{y}^{:h})) - \mathbf{y}^{-p:}\right]\right\| - \left\|\mu(f_{\text{uni}}^{*}(\mathbf{y}^{:h})) - \mathbf{y}^{-p:}\right]\right\|\right]$, where $f_{\text{uni}}^{*}$ is the performance of the best univariate model. Because none of the existing state-of-the-art models show little performance difference between univariate and multivariate forecasting, we also introduce the variate counterfactual scenario, which also performs prediction also using $x^{-p:}$, or the future values of the correlate variates. Formally, this is represented with: $\mathbb{E}_{\mathbf{x}^{:h},\mathbf{x}^{-p:},\mathbf{y}^{:h},\mathbf{y}^{-p:}\sim\mathcal{D}_{\text{eval}}}\left[\left\|\mu(f_{\text{model}}(\mathbf{x}^{:h},\mathbf{x}^{-p:}\mathbf{y}^{:h})) - \mathbf{y}^{-p:}\right]\right\|\right]$. This setting is successfully used by [56] and gives partial evidence of multivariate capabilities.

## 4 DATASET

To provide context for the data that makes up MuSED-FM, we focus on four primary classes of data, with a brief description of the constituent datasets that comprise these datasets. With 45 different datasets aggregated to form MuSED-FM specific dataset details are reserved for Appendix B, with a summary of dataset information in Table 1. The four categories of data are: Traditional multivariate time series datasets (traditional), related univariate time series combined as time series (combined), sequential data transformed into time series (sequence), and synthetic time series (synthetic). These four categories allow us to distinguish correlate selection as by intuition, heuristics, domain and construction, properties we discuss further below.

### 4.1 TRADITIONAL MULTIVARIATE TIME SERIES

We first form the core of our dataset with an aggregation of **37 different datasets** from a wide range of domains including energy, retail, finance, weather, healthcare, public services, healthcare and web traffic. The obvious power of these datasets is primarily that they are readings from a real world process, but also that they have been selected by domain experts, so that the variates provide useful information for forecasting. However, it is also possible that for some of these datasets, even though the variates are intuitively related, they provide no additional forecasting benefit. Thus, we provide analysis in Section 5 to provide initial evidence that these variates should help in forecasting. We were able to collect ~650 time series comprising 0.5 Billion time points.

### 4.2 COLLECTIONS OF UNIVARIATE TIME SERIES

In addition to hand-curated multivariate time series, correlated univariate series, especially in domains such as web and finance, offers another rich data source. MuSED-FM includes **two domains**: stocks and wikipedia pageview data, constructing correlated variates using domain specific heuristics (see Appendix C). The degree of multivariate capability is decided by the efficacy of the heuristic, though we also provide metrics that suggest that the variates are useful for forecasting in Section 5. These collections add ~1800 time series and 29 million data points, as seen in Table 1.

### 4.3 SEQUENTIAL DATA TRANSFORMED INTO TIME SERIES

Time series are a ubiquitous form of data, represented by any signal taken at timestamped intervals. However, most datasets focus on "classic" time series domains such as climate, web, finance, etc. However, sequential data can often be extracted from sources such as scientific data, sequential decision making data, images, language, and video, which we leverage in MuSED-FM, which we leverage to add **five additional** data-rich and correlated domains. With each of these domains, we apply a three step process to convert them into time series: **1)** transform the data into high dimensional tokens, **2)** Apply dimensionality reduction to a fixed set of variates. **3)** Apply smoothing and/or noising to better match the distribution of traditional time series. Specific details for this process are included in Appendix D. The two advantages over other multivariate time series datasets come through the volume of data, as well as high confidence of correlation in the variates, since they are generated according to the same underlying generative process.

| Group | Statistic | Description |
|---|---|---|
| Univariate Basic | Outlier percentage | Percentage of data points $> 3$ st. dev. from mean |
| | Kurtosis | Fourth order moment, measures tailedness |
| | Skewness | Third order moment, measures asymmetry |
| | Entropy [41] | Measures noise as $-\sum_i p_i \log p_i$ |
| | Hurst coefficient [60] | Measure of persistent features |
| | Number of mean crossing points [83] | Number of times the mean is crossed (variance) |
| | Unitroot KPSS statistic [66] | Kwiatkowski-Phillips-Schmidt-Shin measure of stationarity |
| | Unitroot Phillips–Perron statistic [95] | Phillips-Perron test statistic for stationarity. |
| Univariate Shape | Autocorrelation at lag 10 [10] | Measure of the correlation between a data point and another point that occurred 10 time periods earlier |
| | Differenced autocorrelation at lag 1 [10] | Degree to which autocorrelation explains the sequence |
| | Sum of squares of the first five autocorrelation coefficients [10] | Magnitude of simple autocorrelation |
| | Longest flat sequence [83] | length of flat periods, flatter is sometimes easier to predict. |
| | Normalized number of periods [41] | More periodic signals can be easier to predict. |
| | Nonlinearity via surrogate data [123] | measure of linearity in the signal. |
| Univariate Window | Stability [14] | variance of the means in the windows |
| | Lumpiness [14] | variance of the variances in the windows |
| | Max change per window [14] | largest delta inside a window (measures spikes) |
| | Max variance per window [14] | Highest window variance (measures noisiness) |
| Univariate Decomp. | 10-feature Fourier decomposition residual [10] | Residual after decomposing the top 10 frequencies |
| | 10-feature SINDy decomposition residual [13] | Residual after dynamic system decomposition with 10 parameters. |
| | 10-feature Spectral decomposition residual [41] | Residual after decomposition using spectral Principle Component Analysis (linear). |
| | Trend [27] | Measure of trend of STL relative to residual. |
| | Seasonal strength [27] | Measure of seasonal of STL relative to residual. |
| | Spike [83] | Measure of variance of residual. |
| | Linearity [83] | linear consistency of the trend component. |
| | Curvature [83] | Measures curvature of trend component |
| | Peak [83] | measures the peaks of the seasonal component |
| | Trough [83] | Measures the troughs of the seasonal component. |
| | ST decomposition residual [27] | Measures the residual from just the trend and seasonal components. |
| Multivariate | Lagged cross correlation [10] | Convolves the shape of a lagged historical window of a correlate with the future |
| | Granger causality [52] | Measures the degree to which linear model of each variate improves over univariate autoregression. |
| | Transfer entropy [109] | Measures the relative information added by adding a variate. |
| | Convergent cross mapping [115] | Measures the degree of coupling of the variates to the target forecast. |
| | DLinear [142] | Learns a model of the STL decomposition of variates to forecast the target. |

Table 2: **Description of univariate and multivariate predictability features.** Our novel univariate and multivariate predictability metrics are obtained by aggregating these individual metrics.

## 4.4 SYNTHETIC MULTIVARIATE TIME SERIES DATA

Training tabular and time series with synthetic data has seen recent success [102; 56; 6; 5], suggesting that synthetic data provides a substantial breadth of multivariate data that can overlap the space of real time series. Furthermore, synthetic datasets are multivariate by design, since the generative process dictates correlation in time and across variates. MuSED-FM leverages **two distinct lines of work** to generate multivariate time series. First, building on the success of TabPFN [56], especially when applied to time series, we provide an SCM-based implementation building on TabICL [102], but adapted to generate multivariate time series by replacing noise with sequentially correlated inputs, and shifting observations to ensure multivariate predictability. Second, building on the exciting work of Panda [68], we augment this method to construct multivariate dynamical systems with multiple components by composing these systems into arbitrary graph structures, to produce relationships between the graph components that form the multivariate relationships that a foundation model can identify. This allows us to generate 2.5 million time series with 6.2 billion data points, a vast and diverse set of multivariate data with correlations by construction. We provide additional details about these data formats in Appendix E and F.

| | Traditional | | | | | | | | Collections | | Sequential | | | | Synthetic | | Combined |
|---|---|---|---|---|---|---|---|---|---|---|---|---|---|---|---|---|---|
| Predictability Type | Energy | Public Info | Health | Sales | Climate & Env. | Eng. | Finance | Web | Stock | Wiki-pedia | Images | Text | Video | Scien-tific | Dyna-mics | Causal Model | |
| Univariate Pre-dictability | −0.233 | −0.300 | −0.328 | −0.314 | −0.363 | −0.379 | −0.162 | −0.168 | −0.198 | −0.409 | −0.422 | −0.399 | −0.427 | −0.445 | −0.311 | −0.364 | −0.285 |
| Multivariate Predictability | 0.384 | 0.373 | 0.331 | 0.408 | 0.258 | 0.359 | 0.440 | 0.494 | 0.336 | 0.286 | 0.335 | 0.342 | 0.439 | 0.351 | 0.281 | 0.266 | 0.367 |

Table 4: **Univariate and Multivariate predictability metrics per domain.** Here, we showcase both predictability metrics for 16 different domains in the MuSED-FM benchmark. Higher univariate predictability (lower negative values) indicates the availability of more information in the target time series history to forecast effectively. Similarly, higher multivariate predictability indicates the availability of more information in the variates to forecast effectively. **Key takeaway:** across many domains, covariates carry substantial predictive signal that good TSFMs should be able to exploit—beyond simple extrapolation from "single context", i.e., just using single variate.

# 5 ANALYSIS OF CHARACTERISTIC FEATURES

The vital property of a dataset used for time series foundation models is to characterize the degree to which the dataset can be forecasted from historical data. This work introduces two "**predictability**" metrics: **univariate predictability** and **multivariate predictability**. The former measures the degree of difficulty of univariate forecasting, whereas the latter characterizes the degree of simple correlations in the data. Though neither is a perfect metric, this analysis is invaluable for aggregating a large corpus of datasets across a wide variety of different domains, since it allows for a meaningful comparison of foundation model performance in Section 6 across different approximations of difficulty. Both metrics are a averaged sum of different statistics described in Table 2, where the weighting ensures that each statistic can take on a value between [0,1] or [-1,0]. Thus, the metrics take on values between [-1,1] for univariate predictability, and [0,1] for multivariate predictability.

Formally, to construct both the univariate and multivariate "predictability" statistic, we combine these metrics by first normalizing unnormalized values between $0, 1$ using the range observed across all datasets. Next, we take a weighted sum of the metrics, where the weights are $-1$ for metrics that indicate more challenging forecasting when lower, and $+1$ for the opposite (for multivariate, this is when the metric indicates greater information transfer). Finally, we divide by the total number of components to identify the final predictability scores. Thus, for a set of $n$ positive normalized metric values $\{\gamma_1^+, \ldots, \gamma_n^+\}$ and $m$ negative normalized metric values $\{\gamma_1^- \ldots \gamma_m^-\}$, the aggregated metric is computed as:

| | Random | Other | Correlated |
|---|---|---|---|
| **Multivariate Predictability** | 0.163 | 0.135 | 0.367 |

$$\frac{1}{n} \cdot \left( \sum_{i=1}^{n} \gamma_i^+ + \sum_{j=1}^{m} \gamma_j^- \right). \quad (1)$$

Table 3: **Multivariate predictability captures cross-variable signal.** We ablate the metric by replacing true covariates with (i) **Random**: randomly generated Fourier/ARIMA series, and (ii) **Other**: covariates randomly drawn from different series. In both cases, the score drops substantially relative to using the true covariates, validating that the metric assigns higher values when covariates are genuinely correlated and lower values when they are uninformative.

We illustrate these scores across different domains in Table 4.

We visualize higher univariate predictability values indicating greater predictability and thus lower average MAPE across models through a loose correlation as seen in Figure 3a. While multivariate predictability should give a measure of the benefit of adding correlated information, we have observed that most existing multivariate time series foundation models have the same or worse performance as univariate, thus we do not observe that higher multivariate prediction score actually correlate with greater difference in univariate-multivariate in Figure 3b. However, to demonstrate that this is a limitation of the models and not the metric, we construct multivariate time series where we either sample series from a random distribution of Fourier and ARIMA samples, or from unrelated time series, and show that these collections of time series have lower average multivariate predictability than the datasets sampled from MuSED-FM, as seen in Table 3.

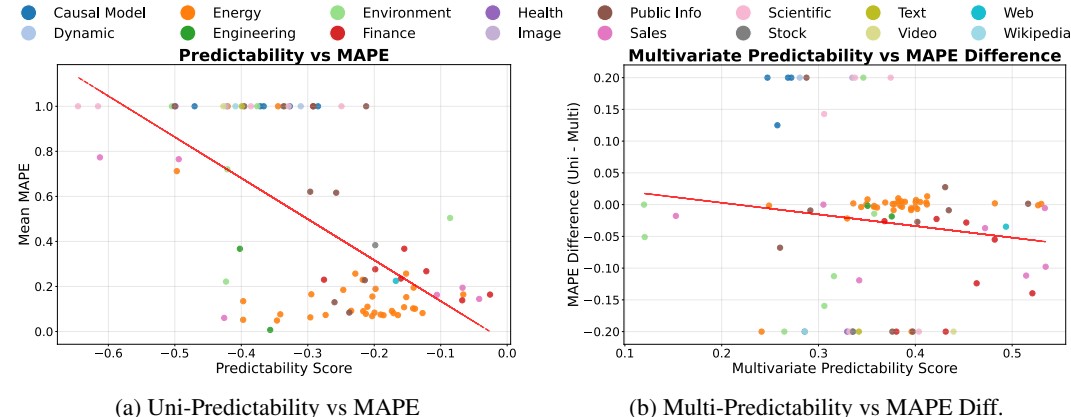

(a) Uni-Predictability vs MAPE      (b) Multi-Predictability vs MAPE Diff.

Figure 3: Points are datasets. **Left:** As predictability increases, MAPE decreases (better prediction from FMs). **Right:** Plot univariate MAPE - multivarate MAPE averaged across multivariate models. **Key takeaway :** As multivariate predictability increases, it does not necessarily indicate better performance of TSFMs, indicating their limited multivariate capability. MAPE values are clipped at 1.0/0.2, respectively, to prevent outlier skew.

| Group | Time Series Model | Description |
|---|---|---|
| Classic | Mean [10] | Trivial Baseline Model that forecasts the mean value. |
| | Prophet [122] | Additive linear regression mixture model using STL. |
| | SARIMAX [10] | Seasonal ARIMA with exogenous regressors; classic statistical model with exogenous elements included as lagged regressors. |
| | Google Causal Impact [12] | Bayesian structural time series model for inferring the causal effects (interventions, counterfactuals, etc.). |
| Univariate | Chronos [5] | Transformer-based large-scale pretrained univariate Time Series Foundation Model. |
| Multivariate | TabPFN [56] | Tabular Transformer-based Model with row-column attention adapted for time series by including timestamps as features trained on synthetic tabular data. Univariate uses only these features, while multivariate uses the historical variates as context. |
| | Mitra [6] | Tabular Transformer-based Model inspired by TabPFN, which uses a similar structure but different implementation and improved performance. Univariate and Multivariate implementations are similar. |
| | Toto [29] | Time-series Optimal Transport Operator; Multivariate time series model trained on proprietary data, with native univariate and multivariate support by changing the number of input columns. |
| | TimesFM [105] | Current state of the art model for univariate time series foundation models. Implements multivariate prediction for the target by first predicting other variates followed by fitting a linear regression model for the target prediction. |

Table 5: **Representative time-series models used in our experiments**, grouped as classic statistical baselines, univariate TSFMs, and multivariate/tabular approaches. The list emphasizes how multivariate modeling is handled: some methods natively learn cross-variable structure (e.g., Toto), while others treat covariates as features or apply post-hoc regression (e.g., TimesFM).

## 6    FOUNDATION MODEL EVALUATION

**Identifying multivariate correlations in real-world time series remains a holy grail of TSFMs**, as it allows the model to maximally leverage the most general form of information—other time series. As such, several state-of-the-art foundation models support multivariate capabilities. However, identifying the degree to which these models are capable of multivariate forecasting remains an open problem. MuSED-FM offers a comprehensive variety of data for identifying multivariate time series foundation model capabilities.

**The core challenge behind evaluating a TSFM's multivariate capability is the entanglement of a *dataset's* multivariate information and a *model's* ability to make use of multivariate infor-**

| | DOMAIN | MEAN | PROPHET | CLASSIC | | CHRONOS | UNIVARIATE | | | | MULTIVARIATE | | | | AGGREGATED |
|---|---|---|---|---|---|---|---|---|---|---|---|---|---|---|
| | | | | SARIMAX | CAUSAL IMPACT | CHRONOS | TABPFN | MITRA | TOTO | TIMESFM | TABPFN | MITRA | TOTO | TIMES FM | |
| TRADITIONAL | ENERGY | N/A | 3.836 | 3.784 | 13.313 | 0.186 | **0.126** | 0.178 | 11.745 | 1.837 | 0.191 | 0.175 | 16.385 | 0.288 | 4.337 |
| | PUBLIC INFO | 2.274 | 1.505 | 2.596 | 8.643 | 1.499 | **1.460** | 3.800 | 1.745 | 1.744 | 1.465 | 3.056 | 1.593 | 1.765 | 2.550 |
| | HEALTH | 5.553 | 4.719 | 1.533 | 2.266 | **0.434** | 2.528 | 3.956 | 0.453 | 3.804 | 5.278 | 4.118 | 0.768 | 2.230 | 2.895 |
| | SALES | 0.487 | 0.524 | 0.539 | 0.355 | 0.242 | 0.247 | 0.455 | 0.240 | 0.217 | 0.476 | 0.464 | 0.265 | **0.213** | 0.363 |
| | CLIMATE & ENVIRONMENT | 2.263 | 1.505 | 1.085 | 0.806 | 2.450 | 1.314 | 2.624 | 1.092 | 0.667 | 1.523 | 2.740 | 1.101 | **0.487** | 1.598 |
| | ENGINEERING | 0.217 | 0.216 | 0.009 | 0.008 | 0.209 | 0.147 | **0.006** | 0.217 | 0.132 | 0.214 | 0.006 | 0.216 | 0.132 | 0.133 |
| | FINANCE | 2.512 | 0.523 | 0.465 | **0.136** | 0.586 | 0.399 | 2.437 | 0.438 | 0.330 | 2.314 | 2.425 | 0.691 | 0.386 | 1.049 |
| | WEB | 0.288 | 0.165 | 0.038 | **0.034** | 0.167 | 0.133 | 0.289 | 0.156 | 0.141 | 0.290 | 0.287 | 0.149 | 0.130 | 0.174 |
| COLLECTIONS | STOCK | 1.132 | 0.268 | **0.026** | 0.027 | 0.251 | 0.274 | 0.136 | 0.202 | 0.200 | 0.885 | 0.129 | 0.204 | 0.199 | 0.303 |
| | WIKIPEDIA | 2.029 | 5.747 | N/A | N/A | **0.462** | 0.584 | 1.348 | 0.551 | 0.802 | 0.481 | 1.282 | 1.583 | 0.802 | 1.425 |
| SEQUENTIAL | IMAGES | 3.590 | 4.245 | 3.346 | 8.042 | 2.114 | 8.958 | 2.373 | 9.727 | 1.871 | 3.560 | 2.259 | 7.447 | **1.752** | 4.560 |
| | TEXT | 2.641 | 3.035 | 1.989 | 1.583 | **1.346** | 1.756 | 2.354 | 1.671 | 1.609 | 2.970 | 2.121 | 1.664 | 1.776 | 2.040 |
| | VIDEO | 1.404 | **0.910** | 2.358 | 1.649 | 0.982 | 1.141 | 2.538 | 1.010 | 1.133 | 1.508 | 2.554 | 0.998 | 1.897 | 1.545 |
| | SCIENTIFIC | N/A | 6.950 | N/A | N/A | 5.522 | 4.816 | N/A | 7.279 | **3.551** | 15.200 | N/A | 5.969 | **3.551** | 6.605 |
| SYNTHETIC | DYNAMIC | 3.023 | 2.796 | 2.792 | 2.887 | 3.325 | 3.081 | 2.204 | 2.675 | 4.393 | **1.094** | 2.219 | 2.298 | 1.610 | 2.646 |
| | CAUSAL MODEL | **1.570** | 16.030 | 10.329 | N/A | 3.753 | 3.922 | 6.883 | 4.434 | 5.496 | 2.113 | 6.430 | 4.408 | 6.489 | 5.988 |
| COMBINED | COMBINED | 2.055 | 3.358 | 2.236 | 3.079 | 1.368 | 2.001 | 2.018 | 2.823 | 1.772 | 2.419 | 1.937 | 2.961 | 1.520 | 2.273 |

Table 6: Domain groups vs forecasting method evaluations mean performance. Green indicates that multivariate performance is more than 0.1 better than the univariate performance of the same model. Note that the best multivariate exceeds the best univariate on only two domains.

**mation.** Poor performance in an evaluation could be attributable to either a dataset's lack of useful information in multivariate signals or a model's inability to take advantage of existing multivariate information. While in an ideal world, we might be confident with a multivariate dataset using a diversity of realistic, meaningful correlates, this volume of data simply does not exist in an open-source format. The core advantage of MuSED-FM is to aggressively leverage a diversity of data as described in Section 4.

**MuSED-FM disentangles the limitations of a model from the limitations of a dataset** by categorizing its forms of data as "synthetic", "combined", or "traditional and sequential". In particular, "synthetic" datasets have multiple multivariate patterns by construction—there will *always* be additional information provided by the variates not present in the univariate time series. Thus, if the model is unable to capture these relationships, then that would serve as a strong indicator of the limitations of the model. On the other hand, "combined" datasets of hand-collated univariate time series will probably contain a good deal of redundant information, so if a model is able to improve performance by adding variates, this suggests generalized ability to ignore certain variates and keep others. Finally, "traditional and sequential" data is likely to contain positive transfer in the correlates, either because of strong human intuitions or because the underlying system from which the sequential information is derived is entangled. Consequently, these domains offer a meaningful blend of being in-distribution of "real" time series data, while in aggregate offering strong evidence . Thus, MuSED-FM is a powerful tool for fine-grained analysis of multivariate time series foundation model capabilities.

**Through MuSED-FM we identify a key limitation of existing TSFMs**– across most domains, these models show **little to no improvement** when presented with multivariate context. For the four models we assessed with multivariate capability (TabPFN, Mitra, Toto, and TimesFM), all of them showed little to no improvement when comparing univariate to multivariate forecasting performance, as seen in Figure 2. Assessing a model univariately involves only showing it the target variate, while multivariate assessment gives the model all the variates. On all domains, the best performance for any model was univariate rather than multivariate, as seen in Table 6. Thus even when a multivariate version of a model is able to improve over the corresponding univariate version, it may not be leveraging the multivariate information to do so.

# 7 CONCLUSION

This work introduces MuSED-FM, a benchmark that assesses time series foundation models multivariate capabilities across a wide range of domains and settings. Focusing on multivariate predictability, this work introduces both univariate and multivariate predictability metrics to estimate useful information from history, as well as from other variables. Combined, this dataset and analysis illustrate a core limitation of existing multivariate time series foundation models: when applied zero-shot on a variety of domains, the *multivariate performance does not exceed univariate performance, even on synthetic datasets which have multivariate correlation by construction*. This demonstrates the need not only for more extensive multivariate analysis, but also the importance of MuSED-FM as a key step for delivering on the promise of multivariate time series foundation models.

## 8 REPRODUCIBILITY STATEMENT

We intend for MuSED-FM as a competition on Huggingface to assess SOTA Multivariate time series foundation models, and eventually fully open source. The analysis computations and metrics leverage open source repositories, and we intend to open source our code for computing the univariate and multivariate predictability metrics.

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

# Appendix

## A    EXTENDED RELATED WORK

Creating meaningful evaluation and training datasets underlies the push for a multivariate time series foundation model, and while main benchmarks remain domain specific, focusing on classic fields such as energy [151; 128], weather and climate [134; 55], traffic [17; 79; 92], finance [88; 58], air quality [147], healthcare and brain signals [32; 62; 126; 96; 136] but also branching out into compute or network related domains [117; 130; 87; 75], cryptocurrency [4], biology [97], industrial devices [125], icicle formation [19], aerospace [144], action identification [49], environment [54]. Alternatively, many benchmarks focus on related problems such as anomaly detection [94; 59; 78; 135; 101], classification [98], robustness [146], etc. The substantial diversity of domains offers both a tantalizing promise and an obvious challenge for time series foundation models: the promise of widespread applicability and the challenge of myriad diversity. Nonetheless, recent work has sought to aggregate and stratify datasets into meaningful benchmarks to assess general time series modeling, including early work such as [84; 119; 85], but subsequently with [3; 47; 9; 1; 71], as well as benchmarks focused on balanced or fair assessment [99; 112], horizon [145], multimodality [76], irregular sampling [15], or dynamical systems [68; 120]. Other work focuses on benchmarking with different foundation models, including [70; 111; 77; 100], while also contributing some compilations of data. While these methods often include multivariate series, they often do not focus on the multivariate problem. First, this can lead to limited volume of multivariate data, and datasets such as those used in [47], [1] and [70] are either entirely or heavily univariate. Furthermore, the analysis of the multivariate datasets in [1] is limited to mostly a small subset.

Of existing time series foundation model datasets, [71] shows the closest resemblance to MuSED-FM, containing multiple real-world time series datasets. Our work subsumes this contemporary work by first including a greater volume of datasets spanning several more domains and time series patterns. Second, including novel datasets by combining related univariate series, transforming sequential data into time series and generating synthetic data to create a more comprehensive datasets, which we discuss further in Section 4. Third, the analysis of variates in [71] uses pearson correlation metrics, which only measure if two series look similar, regardless of how useful the variates are for prediction. By contrast this work introduces a novel multivariate forecasting score that combines Granger Causality [39], transfer entropy [108], convegent cross mapping [138; 116], lagged cross correlation and decomposition-based linear prediction [141], all measures of how the variate can be used for prediction. These tools builds on a rich tradition of time series analysis tools, including collections of metrics such as [83; 14; 2] to modern adaptions [93; 67] to more recent analysis of these tools from a large model perspective [137; 61; 11]. Recent work has investigated other learning metrics for multivariate predictability through dynamics [28; 106], Granger causality [22; 121; 23; 129; 26; 25; 24], causal graphs [82; 20; 21], dynamic bayes networks [118] and transfer entropy [148]. While this work just applies concepts from this field, the analysis applies these concepts on a much larger scale multivariate time series dataset than previously done.

In this work, we add a substantial volume of data in the form of synthetic data and also data derived from classically non-time-series sources: images, video and language. This method for deriving time series follows from understanding time series as timestamped, noisy measurements of a complex, real, dynamic system. From that perspective, time series have been derived from video in Human computer interface [8], video QoE [35], video prediction [143], Sattelite images [114], image compression [33], masked autoencoding [18], time series as images [69], Language [64], Time-series derived from video [72]. Despite strong historic ties—especially considering that many classical methods for analyzing images and video such as FFT and wavelet analysis are derived from time series, this work is the first that we know of applying deep learning transformations to generate multivariate time series from these data sources for time series foundation models, a notion which has seen some recent traction [150]. Furthermore, synthetic data is also a classic tool for the training and evaluation of time series [80], with recent support for synthetic data as an effective training tool [56; 6; 68]. This work includes novel implementations built on [56] and [68] to generate a substantial synthetic evaluation dataset which allows the assessment of a wide range of multivariate relationships which are known to be present by construction.

## B SUMMARY OF DATA PER DATASET

This section provides per-dataset visualizations of time series, evaluation performance and model forecasts. Additional per-data data summaries and example data from each dataset can be found in the supplementary attachment.

| DOMAIN | DATASET | CLASSIC MEAN | PROPHET | SARIMAX | CAUSAL IMPACT | UNIVARIATE CHRONOS | TabPFN | MITRA | TOTO | TIMESFM | MULTIVARIATE TabPFN | MITRA | TOTO | TIMES FM |
|---|---|---|---|---|---|---|---|---|---|---|---|---|---|---|
| ENERGY | EUROPE ELECTRICITY | 0.117 ± 0.046 | 0.069 ± 0.035 | 0.108 ± 0.067 | 0.078 ± 0.033 | 0.064 ± 0.040 | 0.058 ± 0.029 | 0.059 ± 0.032 | 0.076 ± 0.049 | 0.057 ± 0.031 | 0.062 ± 0.037 | 0.060 ± 0.034 | 0.073 ± 0.046 | 0.110 ± 0.087 |
| | DAILY ELECTRICITY | 0.134 ± 0.064 | 0.154 ± 0.082 | 0.122 ± 0.068 | 0.078 ± 0.041 | 0.078 ± 0.045 | 0.083 ± 0.052 | 0.093 ± 0.045 | 0.074 ± 0.040 | 0.082 ± 0.050 | 0.082 ± 0.049 | 0.091 ± 0.039 | 0.074 ± 0.040 | 0.074 ± 0.008 |
| | HOURLY ELECTRICITY | 0.129 ± 0.034 | 0.159 ± 0.136 | 0.213 ± 0.153 | 0.087 ± 0.059 | 0.097 ± 0.067 | 0.107 ± 0.052 | 0.091 ± 0.053 | 0.083 ± 0.046 | 0.093 ± 0.049 | 0.106 ± 0.052 | 0.089 ± 0.049 | 0.084 ± 0.049 | 0.084 ± 0.008 |
| | MDS MICROGRID | 7798.036 ± 489.123 | 124.391 ± 18.456 | 113.663 ± 21.789 | 3.099 ± 0.987 | 0.850 ± 0.104 | 395.502 ± 47.321 | 553.344 ± 58.210 | 57.008 ± 9.876 | 0.713 ± 0.089 | 0.870 ± 0.115 | 553.344 ± 58.210 | 5.809 ± 1.034 | 5.809 ± 0.581 |
| | ERCOT LOAD | 0.125 ± 0.021 | 0.088 ± 0.015 | 0.182 ± 0.025 | 0.069 ± 0.010 | 0.078 ± 0.012 | 0.078 ± 0.013 | 0.081 ± 0.014 | 0.081 ± 0.014 | 0.117 ± 0.018 | 0.119 ± 0.019 | 0.076 ± 0.012 | 0.077 ± 0.011 | 0.077 ± 0.008 |
| | SOLAR ENERGY | 2.722 ± 0.583 | 0.752 ± 0.145 | 6.337 ± 1.026 | 0.144 ± 0.035 | 0.202 ± 0.052 | 0.307 ± 0.061 | 1.253 ± 0.219 | 1.253 ± 0.222 | 2.771 ± 0.558 | 2.770 ± 0.562 | 0.282 ± 0.068 | 1.343 ± 0.276 | 1.343 ± 0.134 |
| | AUSTRALIA ENERGY | 0.255 ± 0.035 | 0.247 ± 0.023 | 0.139 ± 0.108 | 0.118 ± 0.029 | 0.133 ± 0.031 | 0.171 ± 0.036 | 0.158 ± 0.030 | 0.141 ± 0.027 | 0.178 ± 0.031 | 0.146 ± 0.027 | 0.149 ± 0.029 | 0.136 ± 0.026 | 0.136 ± 0.014 |
| | SPAIN ENERGY | 0.153 ± 0.025 | 0.085 ± 0.018 | 0.100 ± 0.015 | 0.070 ± 0.012 | 0.052 ± 0.010 | 0.069 ± 0.011 | 0.082 ± 0.014 | 0.025 ± 0.007 | 0.061 ± 0.011 | 0.068 ± 0.012 | 0.079 ± 0.014 | 0.012 ± 0.004 | 0.012 ± 0.001 |
| CLIMATE & ENVIRONMENT | CAUSAL RIVERS | 2.263 ± 0.350 | 1.505 ± 0.220 | 1.085 ± 0.170 | 0.806 ± 0.120 | 2.450 ± 0.380 | 1.314 ± 0.210 | 2.624 ± 0.400 | 1.092 ± 0.180 | 0.667 ± 0.110 | 1.523 ± 0.250 | 2.740 ± 0.360 | 1.101 ± 0.170 | 0.487 ± 0.048 |
| | OIKOLAB WEATHER | 0.367 ± 0.045 | 0.197 ± 0.030 | 0.422 ± 0.051 | 0.219 ± 0.033 | 0.215 ± 0.034 | 0.316 ± 0.048 | 0.214 ± 0.032 | 0.147 ± 0.027 | 0.318 ± 0.049 | 0.318 ± 0.049 | 0.220 ± 0.035 | 0.095 ± 0.021 | 0.095 ± 0.010 |
| | BEIJING EMBASSY | 2.152 ± 0.432 | 2.235 ± 0.410 | 2.757 ± 0.498 | 3.853 ± 0.475 | 1.877 ± 0.376 | 2.922 ± 0.487 | 1.768 ± 0.354 | 0.772 ± 0.153 | 3.109 ± 0.512 | 2.952 ± 0.491 | 1.804 ± 0.361 | 0.456 ± 0.100 | 0.456 ± 0.046 |
| | BEIJING AQ | 3.173 ± 0.512 | 2.083 ± 0.402 | 2.812 ± 0.534 | 6.472 ± 0.645 | 2.873 ± 0.457 | 3.322 ± 0.498 | 1.934 ± 0.376 | 1.078 ± 0.185 | 2.174 ± 0.410 | 3.329 ± 0.500 | 1.990 ± 0.388 | 0.613 ± 0.131 | 0.613 ± 0.061 |
| | OPEN AQ | 4.797 ± 1.025 | 0.202 ± 0.050 | 2.738 ± 0.503 | 1.261 ± 0.291 | 1.162 ± 0.228 | 6.006 ± 1.083 | 1.054 ± 0.211 | 1.058 ± 0.212 | 1.234 ± 0.241 | 6.608 ± 1.211 | 0.998 ± 0.202 | 0.890 ± 0.165 | 0.890 ± 0.089 |
| ENGINEERING | GAS SENSOR | 0.541 ± 0.085 | 0.324 ± 0.050 | 0.748 ± 0.090 | 0.443 ± 0.065 | 0.444 ± 0.066 | 0.553 ± 0.075 | 0.491 ± 0.070 | 0.541 ± 0.085 | 0.780 ± 0.090 | 0.495 ± 0.070 | 0.494 ± 0.070 | 0.465 ± 0.062 | 0.465 ± 0.046 |
| | EV SENSORS | 0.417 ± 0.065 | 0.426 ± 0.050 | 0.412 ± 0.078 | 0.413 ± 0.060 | 0.290 ± 0.045 | 0.429 ± 0.065 | 0.429 ± 0.065 | 0.260 ± 0.042 | 0.417 ± 0.065 | 0.260 ± 0.043 | 0.427 ± 0.060 | 0.191 ± 0.030 | 0.191 ± 0.019 |
| | VOIP | 0.018 ± 0.007 | 0.007 ± 0.003 | 0.007 ± 0.005 | 0.006 ± 0.004 | 0.005 ± 0.002 | 0.006 ± 0.003 | 0.005 ± 0.003 | 0.004 ± 0.002 | 0.011 ± 0.004 | 0.006 ± 0.003 | 0.005 ± 0.003 | 0.004 ± 0.001 | 0.004 ± 0.001 |
| FINANCE | FRED-MD | 2.512 ± 0.500 | 0.523 ± 0.080 | 0.465 ± 0.073 | 0.586 ± 0.088 | 0.399 ± 0.055 | 2.437 ± 0.512 | 0.438 ± 0.058 | 0.330 ± 0.052 | 2.314 ± 0.485 | 2.425 ± 0.507 | 0.691 ± 0.079 | 0.386 ± 0.065 | 0.386 ± 0.039 |
| | BITCOIN PRICES | 0.677 ± 0.150 | 1.590 ± 0.280 | 1.997 ± 0.320 | 0.540 ± 0.090 | 0.604 ± 0.092 | 0.571 ± 0.084 | 0.481 ± 0.075 | 0.443 ± 0.060 | 0.944 ± 0.092 | 0.606 ± 0.085 | 0.601 ± 0.085 | 0.425 ± 0.070 | 0.425 ± 0.043 |
| HEALTH | CGM | 0.394 ± 0.151 | 1.106 ± 0.783 | 0.571 ± 0.280 | 0.423 ± 0.142 | 0.443 ± 0.177 | 0.398 ± 0.162 | 0.415 ± 0.181 | 0.446 ± 0.132 | 0.394 ± 0.152 | 0.399 ± 0.160 | 0.419 ± 0.196 | 0.446 ± 0.132 | 0.432 ± 0.121 |
| | SLEEP LAB | 0.425 ± 2.149 | 8.331 ± 38.380 | 2.496 ± 10.067 | 4.109 ± 21.275 | 0.425 ± 2.149 | 4.662 ± 29.700 | 7.821 ± 36.355 | 0.462 ± 2.356 | 2.230 ± 12.887 | 5.018 ± 30.028 | 7.492 ± 35.212 | 1.091 ± 5.127 | 10.673 ± 48.438 |
| PUBLIC INFO | TAC | 3.350 ± 3.211 | 9.566 ± 13.125 | 10.030 ± 17.290 | 67.281 ± 122.854 | 8.071 ± 8.225 | 3.390 ± 3.080 | 12.207 ± 16.448 | 7.365 ± 10.528 | 9.662 ± 9.482 | 3.390 ± 3.080 | 19.301 ± 28.149 | 9.153 ± 15.729 | 12.000 ± 20.000 |
| | MN INTERSTATE | 1.438 ± 12.636 | 3.472 ± 0.000 | 3.677 ± 17.904 | 11.859 ± 20.000 | 3.146 ± 13.223 | 1.475 ± 12.888 | 1.805 ± 15.727 | 3.693 ± 15.715 | 3.472 ± 15.618 | 1.476 ± 12.906 | 1.833 ± 15.855 | 3.551 ± 15.567 | 4.200 ± 18.000 |
| | MTA RIDERSHIP | 0.306 ± 0.049 | 0.213 ± 0.036 | 0.360 ± 0.114 | 0.194 ± 0.297 | 0.175 ± 0.040 | 0.330 ± 0.130 | 0.268 ± 0.030 | 0.178 ± 0.035 | 0.177 ± 0.068 | 0.216 ± 0.132 | 0.277 ± 0.038 | 0.175 ± 0.036 | 0.175 ± 0.070 |
| | PARIS MOBILITY | 0.117 ± 0.017 | 0.062 ± 0.023 | 0.141 ± 0.058 | 0.110 ± 0.010 | 0.059 ± 0.021 | 0.086 ± 0.013 | 0.105 ± 0.009 | 0.048 ± 0.012 | 0.045 ± 0.018 | 0.046 ± 0.012 | 0.108 ± 0.014 | 0.048 ± 0.012 | 0.047 ± 0.019 |
| | RIDESHARE | 0.031 ± 0.015 | 0.033 ± 0.016 | 1.281 ± 0.006 | 0.033 ± 0.016 | 0.988 ± 0.074 | 0.222 ± 0.119 | 0.401 ± 0.134 | 0.981 ± 0.013 | 0.570 ± 0.131 | 0.252 ± 0.145 | 0.401 ± 0.134 | 0.989 ± 0.014 | 0.570 ± 0.131 |
| | BLUE BIKES | 0.119 ± 0.040 | 0.434 ± 0.270 | 4.941 ± 4.731 | 2.987 ± 0.861 | 0.434 ± 0.270 | 2.745 ± 0.924 | 7.733 ± 3.423 | 0.808 ± 0.305 | 0.510 ± 0.318 | 0.439 ± 0.236 | 7.956 ± 3.757 | 0.682 ± 0.302 | 0.522 ± 0.297 |
| | AUSTIN WATER | 0.076 ± 0.014 | 0.095 ± 0.035 | 0.233 ± 0.034 | 0.135 ± 0.068 | 0.119 ± 0.036 | 0.093 ± 0.001 | 0.094 ± 0.011 | 0.087 ± 0.011 | 0.089 ± 0.022 | 0.087 ± 0.033 | 0.094 ± 0.011 | 0.099 ± 0.004 | 0.089 ± 0.022 |
| | TRAFFIC | 0.111 ± 0.176 | 0.141 ± 0.169 | 1.188 ± 3.395 | 0.379 ± 0.000 | 0.141 ± 0.169 | 0.203 ± 0.262 | 2.791 ± 5.831 | 0.301 ± 0.456 | 0.183 ± 0.219 | 2.085 ± 5.063 | 2.445 ± 5.204 | 2.722 ± 5.815 | 0.533 ± 0.532 |
| | CURSOR TABS | 0.869 ± 0.150 | 1.478 ± 0.200 | 1.606 ± 0.215 | 2.158 ± 0.250 | 2.158 ± 0.250 | 2.158 ± 0.250 | 2.284 ± 0.260 | 2.873 ± 0.295 | 3.419 ± 0.300 | 3.722 ± 0.320 | 3.914 ± 0.325 | 3.972 ± 0.328 | 4.796 ± 0.350 |
| SALES | WALMART SALES | 0.041 ± 0.020 | 0.041 ± 0.021 | 0.045 ± 0.021 | 0.045 ± 0.021 | 0.045 ± 0.021 | 0.049 ± 0.021 | 0.050 ± 0.027 | 0.050 ± 0.027 | 0.057 ± 0.027 | 0.063 ± 0.047 | 0.064 ± 0.026 | 0.065 ± 0.028 | 0.074 ± 0.035 |
| | BLOW MOLDING | 0.077 ± 0.085 | 0.096 ± 0.058 | 0.097 ± 0.073 | 0.097 ± 0.073 | 0.097 ± 0.073 | 0.098 ± 0.078 | 0.101 ± 0.077 | 0.104 ± 0.035 | 0.119 ± 0.063 | 0.119 ± 0.107 | 0.148 ± 0.102 | 0.242 ± 0.079 | 0.195 ± 0.070 |
| | PASTA SALES | 0.527 ± 0.170 | 0.531 ± 0.154 | 0.539 ± 0.174 | 0.575 ± 0.161 | 0.606 ± 0.167 | 0.609 ± 0.340 | 0.613 ± 0.341 | 0.613 ± 0.145 | 0.713 ± 0.690 | 0.877 ± 1.263 | 0.943 ± 0.746 | 1.023 ± 1.438 | 0.900 ± 0.750 |
| | RICE PRICES | 0.066 ± 0.056 | 0.075 ± 0.047 | 0.103 ± 0.114 | 0.104 ± 0.096 | 0.105 ± 0.108 | 0.105 ± 0.104 | 0.107 ± 0.113 | 0.107 ± 0.113 | 0.107 ± 0.115 | 0.115 ± 0.102 | 0.115 ± 0.139 | 0.366 ± 0.120 | 0.396 ± 0.121 |
| | GOLD PRICES | 0.509 ± 0.064 | 0.122 ± 0.072 | 0.074 ± 0.027 | 0.080 ± 0.058 | 0.079 ± 0.034 | 0.073 ± 0.031 | 0.150 ± 0.092 | 0.082 ± 0.024 | 0.074 ± 0.025 | 0.425 ± 0.043 | 0.163 ± 0.079 | 0.091 ± 0.025 | 0.074 ± 0.025 |
| WEB | WEB VISITORS | 0.288 ± 0.049 | 0.165 ± 0.033 | 0.480 ± 0.162 | 0.031 ± 0.011 | 0.167 ± 0.047 | 0.133 ± 0.048 | 0.289 ± 0.062 | 0.156 ± 0.013 | 0.141 ± 0.038 | 0.290 ± 0.057 | 0.287 ± 0.058 | 0.149 ± 0.012 | 0.130 ± 0.048 |
| SCIENTIFIC | MUJOCO | 1.346 ± 0.824 | 1.695 ± 1.269 | 6.115 ± 6.839 | 456.120 ± N/A | 6.393 ± 13.442 | 4.072 ± 5.288 | 2.316 ± 2.522 | 5.197 ± 8.799 | 3.434 ± 3.657 | 4.120 ± 5.194 | 2.408 ± 2.819 | 5.352 ± 9.792 | 3.434 ± 3.657 |
| | SPRITEWORLD | 168.547 ± 49.732 | 27.964 ± 5.128 | 181.612 ± 40.211 | 105.031 ± 29.874 | 2.034 ± 0.982 | 7.796 ± 1.987 | 1597.468 ± 198.645 | 15.609 ± 4.872 | 4.022 ± 1.489 | 59.523 ± 9.874 | 1002.461 ± 149.732 | 8.441 ± 2.471 | 4.023 ± 1.512 |
| IMAGE | CIFAR100 | 3.590 ± 1.124 | 4.245 ± 1.036 | 1.361 ± 0.531 | 8.042 ± 2.145 | 2.114 ± 0.897 | 8.958 ± 2.012 | 2.373 ± 0.784 | 9.727 ± 2.355 | 1.871 ± 1.210 | 3.560 ± 1.142 | 2.259 ± 0.831 | 7.447 ± 2.101 | 1.752 ± 1.241 |
| TEXT | OPENWEBTEXT | 2.641 ± 0.950 | 3.035 ± 1.012 | 1.541 ± 2.806 | 1.583 ± 0.875 | 1.346 ± 0.890 | 1.756 ± 0.912 | 2.354 ± 0.870 | 1.671 ± 0.865 | 1.609 ± 3.599 | 2.970 ± 1.020 | 2.121 ± 0.895 | 1.664 ± 0.900 | 1.897 ± 6.083 |
| VIDEO | KITTI | 1.404 ± 0.950 | 0.910 ± 0.875 | 1.084 ± 3.800 | 1.649 ± 0.870 | 0.982 ± 0.890 | 1.141 ± 0.912 | 2.538 ± 0.895 | 1.010 ± 0.865 | 1.133 ± 3.993 | 1.508 ± 1.020 | 2.554 ± 0.900 | 0.998 ± 0.880 | 1.897 ± 6.083 |
| SYNTHETIC | CAUSAL MODEL | 16.024 ± 3.456 | 10.329 ± 54.528 | 706217.234 ± 0.000 | 3.750 ± 5.591 | 3.813 ± 2.988 | 6.616 ± 4.298 | 4.446 ± 3.561 | 5.496 ± 25.451 | 2.159 ± 0.555 | 6.132 ± 2.563 | 4.372 ± 3.660 | 6.489 ± 26.850 | 7.497 ± 32.990 |
| | DYNAMIC | 3.023 ± 1.500 | 2.796 ± 1.200 | 8.724 ± 29.279 | 2.887 ± 1.000 | 3.325 ± 1.000 | 3.081 ± 1.200 | 2.204 ± 1.100 | 2.675 ± 1.150 | 4.393 ± 11.047 | 1.850 ± 0.900 | 2.220 ± 1.050 | 2.299 ± 1.100 | 1.610 ± 2.959 |
| WIKIPEDIA | WIKIPEDIA | 2.029 ± 0.500 | 5.747 ± 1.000 | 45.000 ± 10.000 | 48.093 ± 15.000 | 0.462 ± 0.080 | 0.584 ± 0.100 | 1.348 ± 0.250 | 0.551 ± 0.090 | 0.802 ± 1.146 | 0.481 ± 0.080 | 1.282 ± 0.200 | 1.583 ± 0.250 | 0.820 ± 1.150 |
| STOCK | NASDAQ TRACKER | 1.782 ± 0.400 | 0.450 ± 0.100 | 0.192 ± 0.194 | 0.035 ± 0.010 | 0.179 ± 0.050 | 0.201 ± 0.050 | 0.208 ± 0.050 | 0.175 ± 0.050 | 0.200 ± 0.201 | 0.029 ± 0.010 | 0.190 ± 0.050 | 0.174 ± 0.050 | 0.199 ± 0.173 |

Table 7: Domain groups vs forecasting method evaluations mean performance with standard deviations.

### B.1 SAMPLES OF DATASET EVALUATIONS

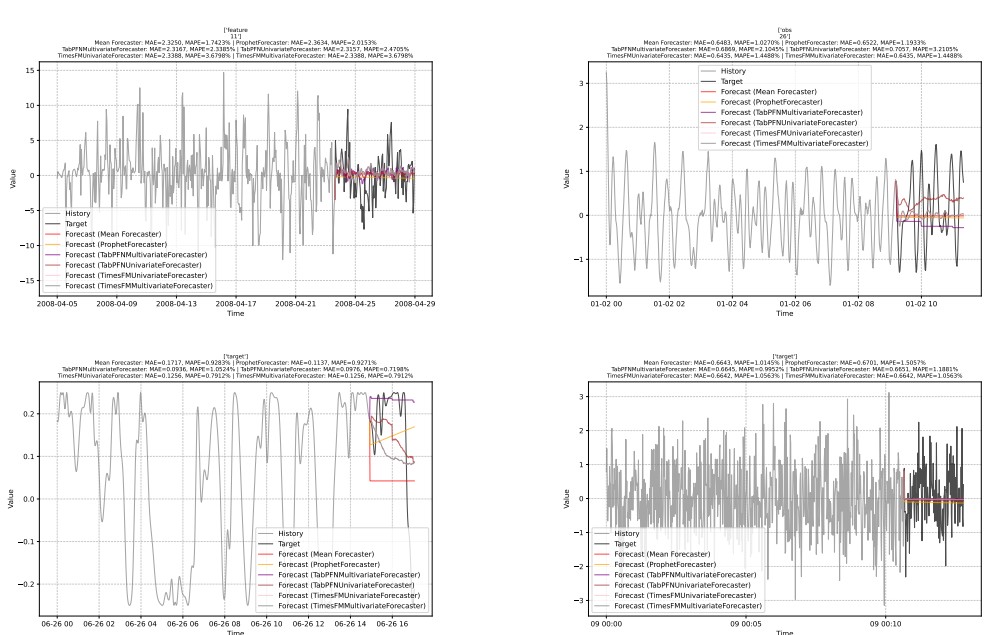

Figure 4: Cross-domain samples I: vision, control, finance, synthetic

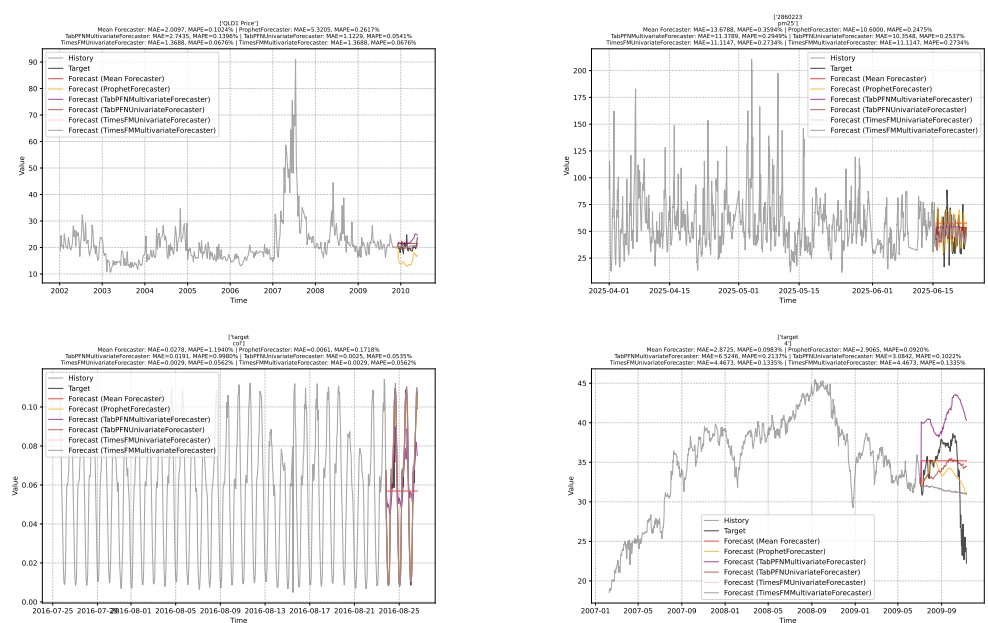

Figure 5: Cross-domain samples II: electricity, air quality, traffic, dynamics

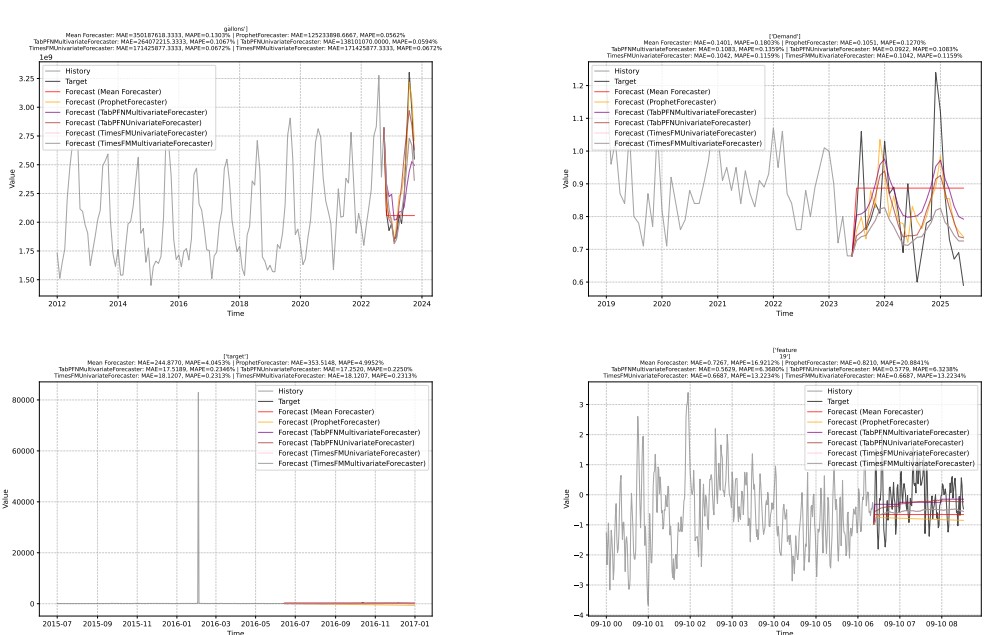

Figure 6: Cross-domain samples III: water, regional electricity, wikipedia, language

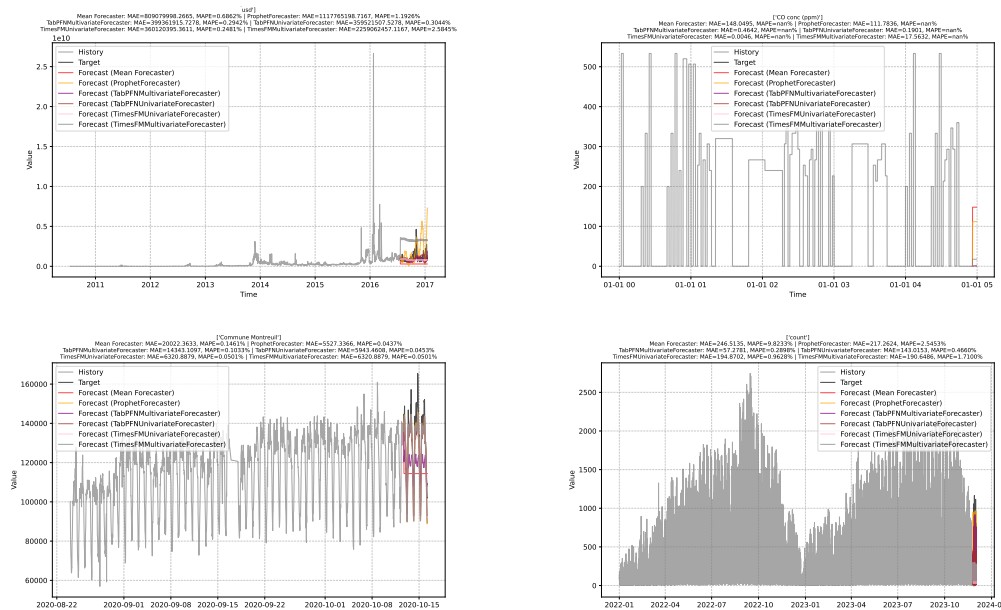

Figure 7: Cross-domain samples IV: Bitcoin price, gas sensors, mobility, bicycle sales

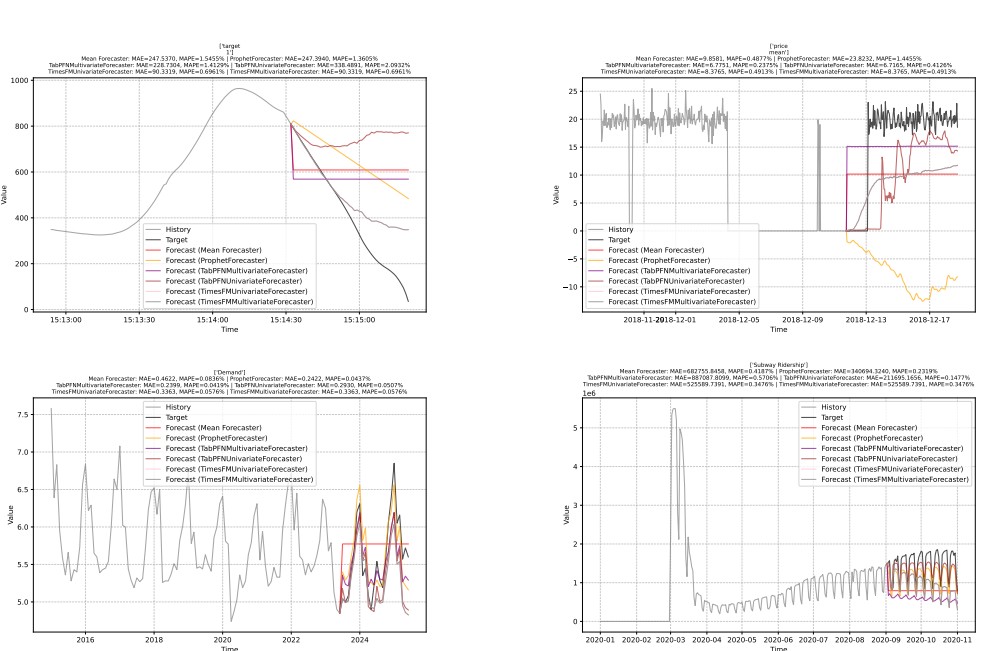

Figure 8: Cross-domain samples V: Video, rideshare, power, transit

## B.2 FULL MULTIVARIATE SERIES

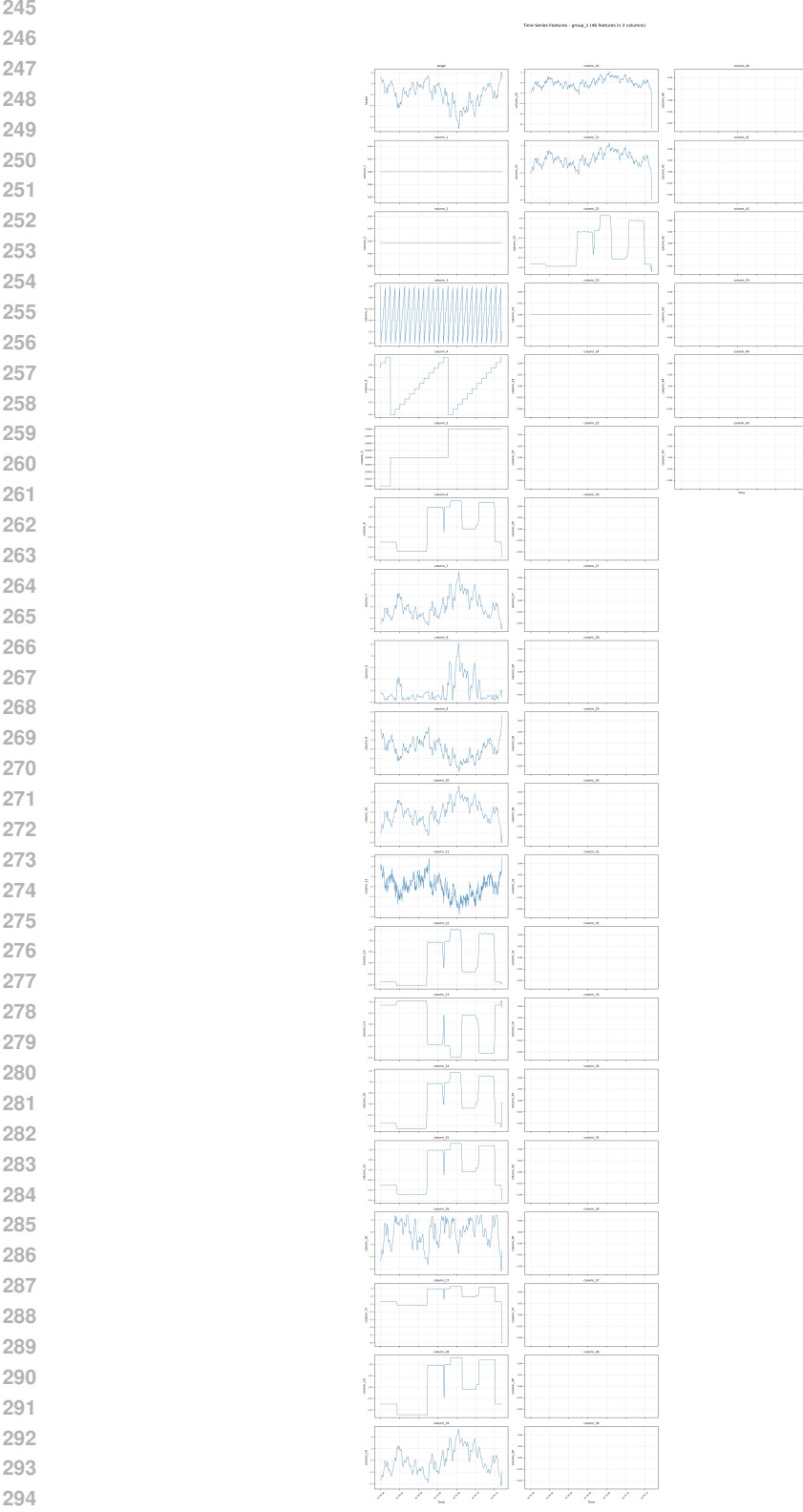

Figure 9: Synthetic Causal Data

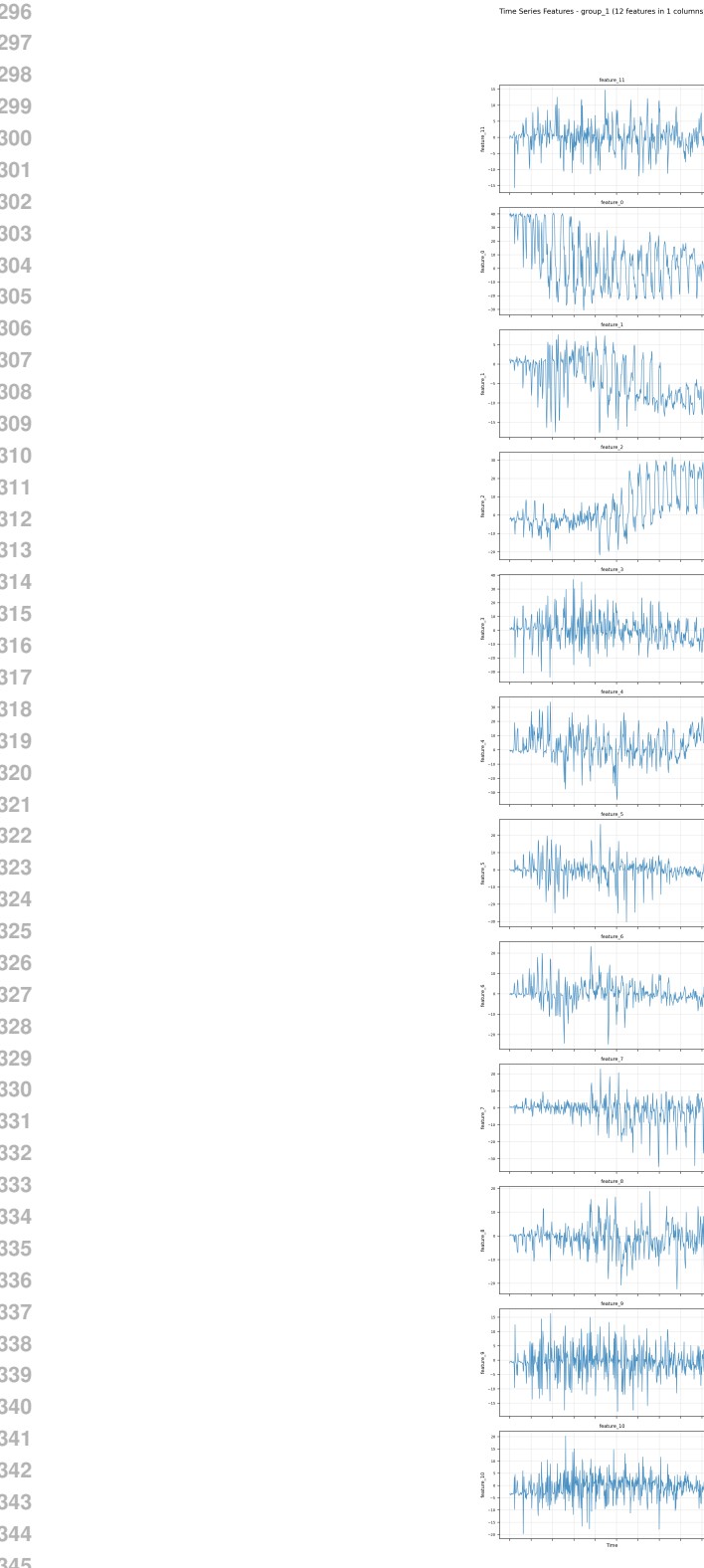

Figure 10: Sequential CIFAR-100 image data

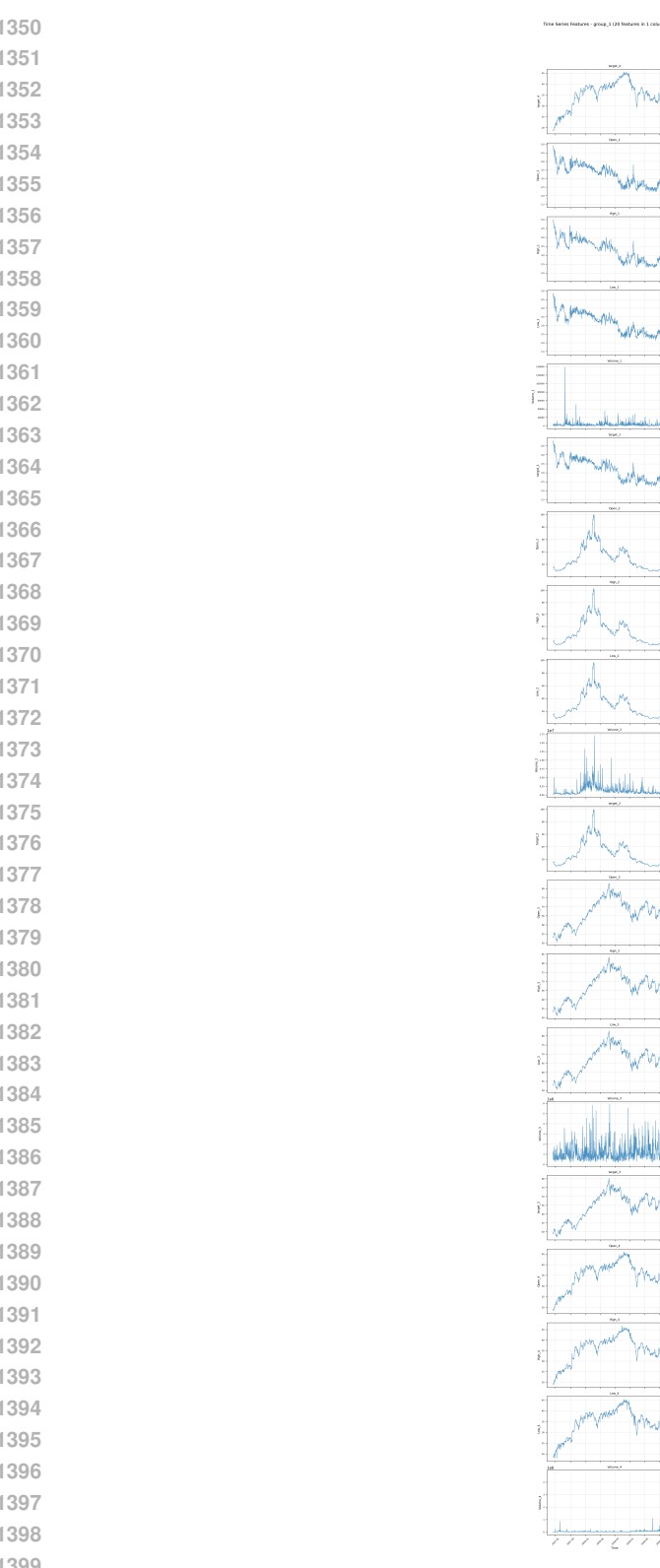

Figure 11: Collected Stock data

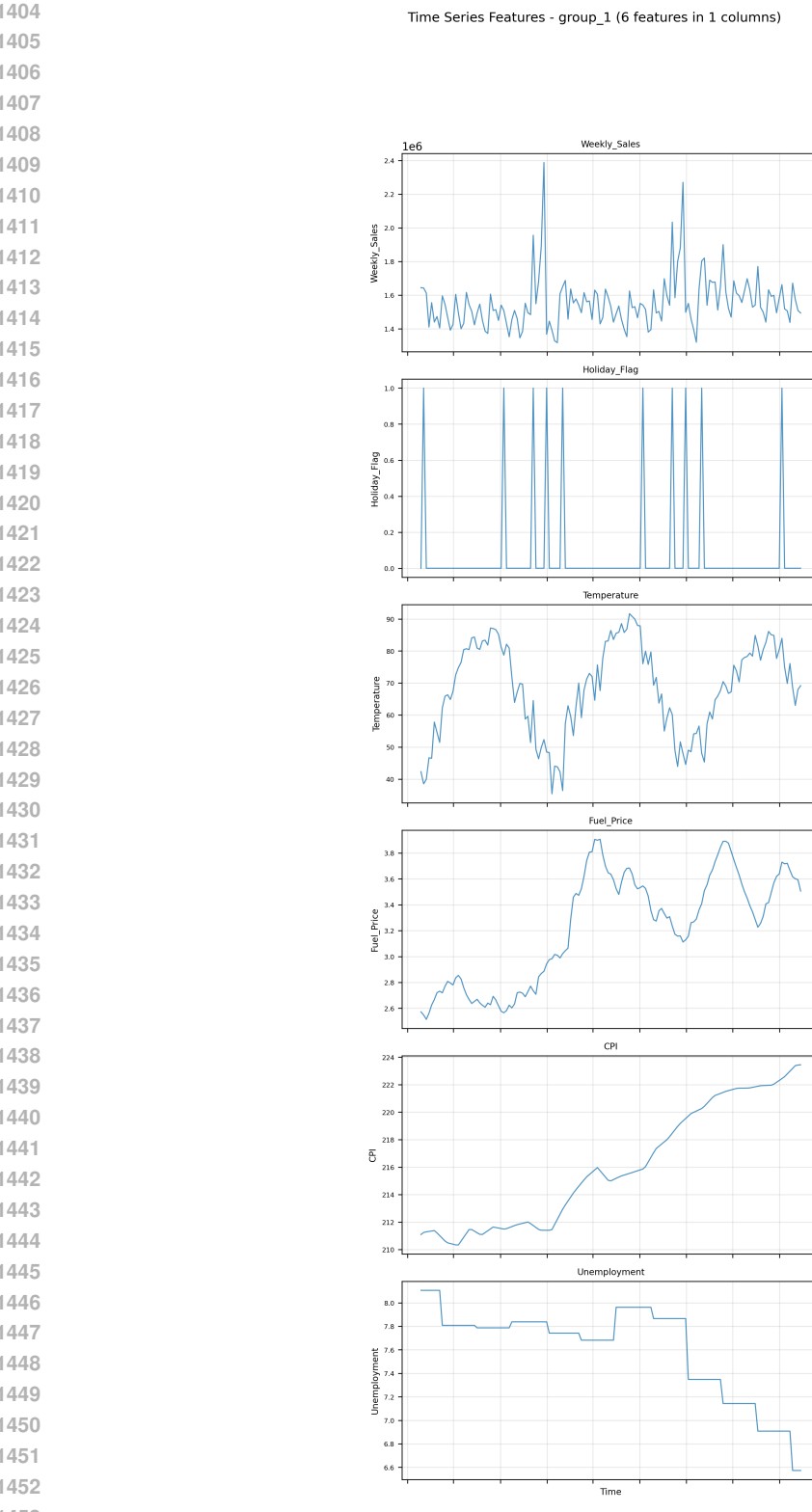

Figure 12: Walmart Sales Dataset

## C COMBINING COLLECTIONS OF RELATED UNIVARIATE TIME SERIES

### C.1 STOCK DATA

We collect historical daily stock price data for the tickers currently trading on the NASDAQ exchange. The list of active tickers is obtained from Nasdaq Trader, and the historical time series are downloaded from Yahoo Finance via the `yfinance` Python package. For each ticker, data are saved as CSV with the following fields

- Date: trading day.
- Open: opening price.
- High: highest price during the day.
- Low: lowest price during the day.
- Adj Close: closing price adjusted for both dividends and splits.
- Volume: the number of shares traded that day.

For our evaluation, we select the period 2016-04-05 → 2020-04-01 and include 2347 tickers. We group companies by type and construct sequences in which each sequence contains five stocks. The prediction target is the **Close** price for one of the stocks, randomly selected.

### C.2 WIKIMEDIA DATA

We collect daily Wikipedia pageview data for 145,000 pages over the period 2015-07-01 → 2016-12-31. For each page, semantic categories are obtained from the Wikipedia API category query interface. Categories that appear in more than 5% of all pages are removed to avoid generic groupings. Each page is then represented as text by concatenating its title with the remaining categories.

We generate embeddings for these text representations using Sentence-BERT [104]. Clustering is performed with HDBSCAN [86] to group pages by semantic similarity.

From the clusters, we construct multivariate time series datasets. Clusters with more than 30 pages are split into groups of 30 pages each. Each group forms a multivariate series with 30 covariates, where each covariate is the daily pageview count for one page. This process yields 1,530 multivariate series in total.

## D SEQUENTIAL DATA TRANSFORMATION PROCESS

Multivariate time series foundation models must learn to identify patterns from the context and leverage them to predict future values. Thus, when evaluating the capability of these models it can be useful to introduce series that are likely to contain varied correlational patterns from various sources. This is the motivation for transforming data from different sequential processes. In this section we describe the process that we apply to transform data from image, video, language, and simulator domains:

1. Transform into a meaningful sequential space using a tokenizer. This can vary for different domains.
   (a) For images, apply spiral patching with a visual transformer [132] using images from CIFAR-100 [65] by resizing the images to $450 \times 450$, then using a patch size of 16 with a stride of 8.
   (b) For scientific data, we took two simulators, Spriteworld [26], which uses a box2d physics simulation backend where the number of objects and their initial positions and velocities are randomized, and Mujoco [124] data collected using policies of varying capability in D4RL [40]. Each episode lasts 1000 time steps, and since the state for these simulators is already low-dimensional, we do not need to apply additional processing
   (c) For Text, we take the openwebtext dataset [48], and apply an open sourcelanguage model [149] to conver the words into vectors.
   (d) For video data, we apply keypoint detection to the KITTI dataset [45], using the keypoint positions as the states over the course of a time series.

2. Apply dimensionality reduction in the form of PCA to the sequencized data to reduce it to a meaningful form. Given the set of tokens, this takes a substantial random sample to apply SVD, then uses the learned matrix for dimensionality reduction. The number of dimensions taken by PCA is tuned to allow for 80-90% of the variance to be explained by the features. For images, this is 12 dimensions, and for text, this is 20 dimensions. Video and Simulator data require no additional dimensionality reduction.

3. Apply smoothing and/or noise, where smoothing is applied in the form of a Gaussian filter, where the window size is tuned based on the noisiness of the data. Noise is re-added from a normal distribution. For images and text, the Gaussian filter used a window size of 5, whereas for D4RL data, we used a window size of 20, though for Spriteworld, we do not apply filtering. We reapply 3% noise, which gives the files similar characteristics to time series from other domains. For video we do not apply either filtering or noise.

### OpenWebText

EAST ST. LOUIS, Ill., Nov. 1 (UPI) — Two women and a man have been charged with abusing two young girls during a sex party at an Indiana hotel.

Federal prosecutors in southern Illinois say Tabitha Robinson of Midlothian, Ill., brought a 3-year-old in her care to the party, the Chicago Tribune reported. Louise Helen Masulla of New Athens, Ill., allegedly brought an 11-year-old.

The two women and William Milligan of Bloomington Ind., were charged with conspiracy to produce child pornography and conspiracy to transport minors to engage in criminal sexual activity.

Investigators say the three met online and then at a Best Western in Terre Haute. They allegedly found pictures on the two women's computers that showed the children engaged in sexual activity.

Shakespeare famously wrote that people are the stuff dreams are built on. For Sacramento, its dreams of Major League Soccer will be built on 244 acres of undeveloped land in the city's downtown Railyards.

The historic site, a pocket of low-lying, swampy land located where the American and Sacramento Rivers meet, was once the western terminus of the Transcontinental Railroad in the 1860s, but has lain dormant since Union Pacific Railroad workers ceased operations in 1995. A proposed effort to redevelop the area in 2007 was foiled by the economic recession that struck the country at the time, but true to the city's indomitable spirit, a new group of investors has resumed the task of bringing life to the area.

### CIFAR-100

### Spriteworld

### KITTI

### MuJoCo

Figure 13: Examples on selected datasets, including OpenWebText [48], CIFAR-100 [65], Spriteworld [26], KITTI [45], and MuJoCo [40].

Altogether, this set of three steps can be applied on a wide range of data. In fact, the datasets we selected were smaller than the general corpus of image, video, language or simulator data, and future work can explore applying this process to generate more multivariate time series data.

## E  STRUCTURAL CAUSAL MODELS FOR SYNTHETIC DATA

The structural causal model strategy for representation learning involves constructing a graph, where each node is a variate and the value at a node is a function of its parents in the graph. Then the root nodes take on values according to some sequentially correlated system (ex. ARIMA model, random fourier features, etc.). At each timestamp the values are then propagated through the graph. A consistent subset of the nodes are used as metadata, and one node is selected as the target variable. The metadata nodes are then lagged in time by a random amount to ensure that the target variable is a function of the history of the metadata.

Formally, the data distribution is dictated by $g_{\text{SCM}}$, constructed in practice using MLPs and loosely derived from [56; 102]. Each dataset is described by a graph, where each node in the graph is a variate. The set of variates that are not caused by other variates are the inputs $s_{\text{in}}$, a set of intermediate nodes, some of which are hidden $s_{\text{hidden}}$, a set of observed nodes $x$ and target nodes $y$, and a set of noise nodes $u_{\text{hidden}}$. An SCM is defined by $g_{\text{SCM}}(\mathcal{S}, \mathcal{U}, \mathcal{E})$, where $\mathcal{E}$ are the set of directed edges arranged in a directed acyclic graph. The input nodes are independently sampled according to some noise distribution $S_{\text{in}}$, the noise nodes are sampled according to $U_{\text{hidden}}$, and the hidden nodes are computed as a function of the parents in the graph, $s_i := g_i(\text{PA}_i(\mathbf{s}), u_i)$. In practice, $g_i$

is represented with nonlinearity activation function $a : \mathbb{R} \to \mathbb{R}$, a weight vector $w^{|\text{PA}|}$ according to $s_i = a(w^\top \text{PA}_i(\mathbf{s})) + u_i$. For this work, we sample the noise correlated in time, where we use the following set of functions:

1. ARMA functions with random AR and MA components
2. Random piecewise linear functions with between 8-20 pieces.
3. Impulse functions where the impulses are sampled at random times, between 8-20 impulses
4. Random fourier features, with between 8-20 frequencies sampled
5. Weiner Distribution with trend component
6. Random piecewise splines, with between 8-20 pieces.

For non-input nodes, the noise parameter is 0.01% Gaussian noise.

The observed nodes and target nodes are selected randomly from the graph, and a lag operator is applied to the observations of the observed node, so that without loss of generality if metadata node $x_i$ is the only parent of $y$, then $y = f(x_i^t)$, thus ensuring a lagged relationship between the target and the metadata node.

# F DYNAMICAL SYSTEM SYNTHETIC DATA (SKEW-PRODUCT GENERATION)

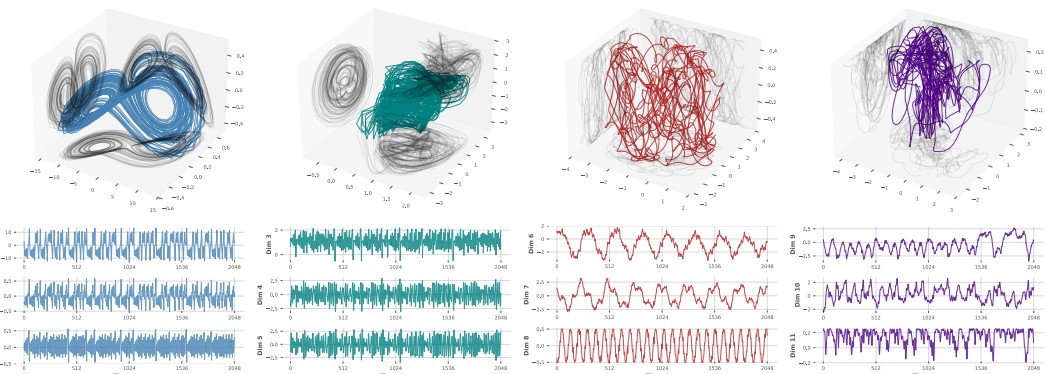

Figure 14: Example skew-product dynamical system after two generations of recombination.

Starting with a founder pool of 129 chaotic dynamical systems from the `dysts` dataset [46], we randomly perturb the parameters of each in a Gaussian ball around the default value. In Generation 1, we sample pairs of perturbed systems (which we denote the driver and the response system) and couple their flows together in a skew-product form. In particular, the coupling is unidirectional and defines the influence of the driver system on the response system. For each candidate perturbed skew system that successfully integrates, we apply a suite of attractor tests to eliminate trajectories that are converging to a fixed point, diverging to infinity, limit cycles, or simple straight lines. We then apply the 0-1 test [37] to eliminate periodic and quasiperiodic systems. We check for a continuous broadband power spectrum, as well as positive leading Lyapunov exponent via the data-driven Rosenstein estimator [107]. Lastly, we check for stationarity via a combination of the Kwiatkowski-Phillips-Schmidt-Shin (KPSS) [66] and Augmented Dickey–Fuller (ADF) [34] tests. In Generation 2, we sample pairs of successful perturbed skew systems and couple them together in the same fashion as before. By construction, we have a combinatorial growth in the number of possible distinct pairings of systems to couple together. This extends the framework of [68] and allows us to create higher-dimensional systems with arbitrary causal relationships. Additional dynamical diversity can be introduced via the choice of coupling maps.

Let $m, n \in \mathbb{N}$ denote the driver and response system dimension, respectively. Suppose we have a driver system $\dot{\mathbf{x}} = \mathbf{f}(\mathbf{x})$ and a response system $\dot{\mathbf{y}} = \mathbf{g}(\mathbf{y})$. A simple coupling strategy is an *Additive Coupling Map* in which we randomize the dimension ordering of the driver system, normalize the flows by their RMS, and then simply add the driver flow to the response flow $\dot{\mathbf{y}} = \mathbf{g}(\mathbf{y}) + \dot{\mathbf{x}}$. Concatenating dimensions, the flow of the cominbed skew product system is simply $[\mathbf{f}(\mathbf{x}), \mathbf{f}(\mathbf{x}) + \mathbf{g}(\mathbf{y})]$, modulo the normalization and randomization of dimension ordering.

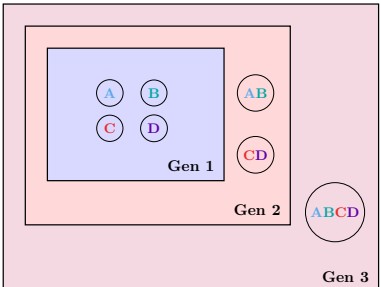

Given a desired rank $r$, we provide the option to construct the skew coupling matrix $C$ as a low-rank positive semidefinite perturbation. Randomizing the dimension ordering of the driver system,

$$C = \begin{bmatrix} D & I_n + VSV^\top \end{bmatrix}, \quad D = \left( I_{\max\{n,m\}} \right)_{1:n, \ 1:(m)},$$

$V \in \mathbb{R}^{n \times r}$ with $V_{ij} \overset{\text{i.i.d.}}{\sim} \mathcal{N}(0,1)$, $\|v_k\|_2 = 1$ (column-normalized). And $S = \text{diag}(s_1, \ldots, s_r), \ s_k \sim \mathcal{U}[0,1]$.

Our *Activated Coupling Map* first applies this coupling matrix to the driver and response flows. As before, we normalize each flow by its RMS, but before adding them together, we pass the transformed driver flow through a nonlinearity (we choose a hyperbolic tangent function).

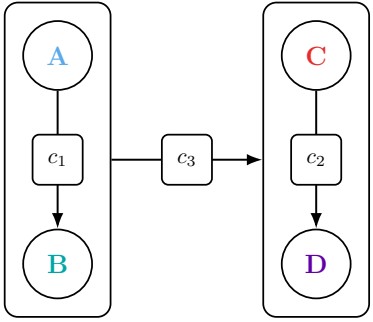

Figure 15: Skew-Product Generation

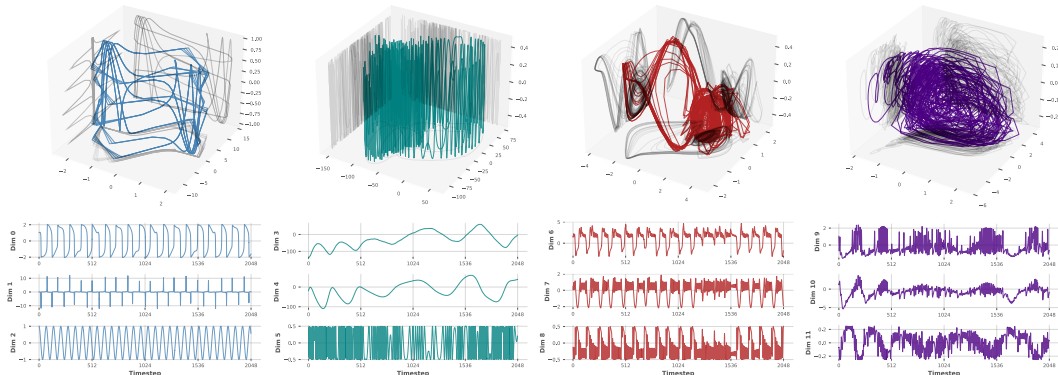

Figure 16: Example skew-product dynamical system after two generations of recombination.

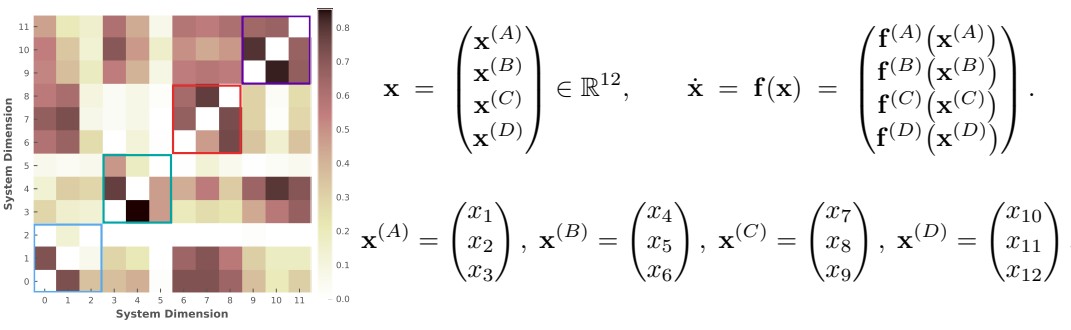

$$\mathbf{x} = \begin{pmatrix} \mathbf{x}^{(A)} \\ \mathbf{x}^{(B)} \\ \mathbf{x}^{(C)} \\ \mathbf{x}^{(D)} \end{pmatrix} \in \mathbb{R}^{12}, \qquad \dot{\mathbf{x}} = \mathbf{f}(\mathbf{x}) = \begin{pmatrix} \mathbf{f}^{(A)}\big(\mathbf{x}^{(A)}\big) \\ \mathbf{f}^{(B)}\big(\mathbf{x}^{(B)}\big) \\ \mathbf{f}^{(C)}\big(\mathbf{x}^{(C)}\big) \\ \mathbf{f}^{(D)}\big(\mathbf{x}^{(D)}\big) \end{pmatrix}.$$

$$\mathbf{x}^{(A)} = \begin{pmatrix} x_1 \\ x_2 \\ x_3 \end{pmatrix}, \ \mathbf{x}^{(B)} = \begin{pmatrix} x_4 \\ x_5 \\ x_6 \end{pmatrix}, \ \mathbf{x}^{(C)} = \begin{pmatrix} x_7 \\ x_8 \\ x_9 \end{pmatrix}, \ \mathbf{x}^{(D)} = \begin{pmatrix} x_{10} \\ x_{11} \\ x_{12} \end{pmatrix}.$$

Figure 17: CCM Map

$$\phi(u) := \tfrac{1}{2}\big(|u+1| - |u-1|\big).$$

Flow RHS of Component Systems

$$\mathbf{f}^{(A)}(\mathbf{x}^{(A)}) = \begin{pmatrix} x_2 \\ \mu\left(1 - x_1^2\right)x_2 - x_1 + a\,\sin(x_3) \\ \omega \end{pmatrix} \qquad \textit{(ForcedVanDerPol)},$$

$$\mathbf{f}^{(B)}(\mathbf{x}^{(B)}) = \begin{pmatrix} a\,x_5 - a\,x_4 + d\,x_4 x_6 \\ k\,x_4 + f\,x_5 - x_4 x_6 \\ c\,x_6 + x_4 x_5 - \varepsilon x_4^2 \end{pmatrix} \qquad \textit{(Tsucs2)},$$

$$\mathbf{f}^{(C)}(\mathbf{x}^{(C)}) = \begin{pmatrix} -x_7 + d\,\phi(x_7) - b\,\phi(x_8) - b\,\phi(x_9) \\ -x_8 - b\,\phi(x_7) + c\,\phi(x_8) - a\,\phi(x_9) \\ -x_9 - b\,\phi(x_7) + a\,\phi(x_8) + \phi(x_9) \end{pmatrix} \qquad \textit{(CellularNeuralNetwork)},$$

$$\mathbf{f}^{(D)}(\mathbf{x}^{(D)}) = \begin{pmatrix} -x_{10} + t_x^{-1}x_{11} - \tfrac{a}{t_x}x_{10}^3 + \tfrac{b}{t_x}x_{10}^2 + t_x^{-1}x_{12} \\ -a\,x_{10}^3 - (d-b)x_{10}^2 + x_{12} \\ -\tfrac{s}{t_z}x_{10} - t_z^{-1}x_{12} + \tfrac{c}{t_z} \end{pmatrix} \qquad \textit{(HindmarshRose)}.$$

System Parameters

$$\begin{aligned} \text{ForcedVanDerPol:} \quad & (a,\,\mu,\,\omega) = (1.2,\,8.53,\,0.63) \\ \text{Tsucs2:} \quad & (a,\,c,\,d,\,\varepsilon,\,f,\,k) = (40,\,0.833,\,0.5,\,0.65,\,20,\,0) \\ \text{CellularNeuralNetwork:} \quad & (a,\,b,\,c,\,d) = (4.4,\,3.21,\,1.1,\,1.24) \\ \text{HindmarshRose:} \quad & (a,\,b,\,c,\,d,\,s,\,t_x,\,t_z) = (0.49,\,1.0,\,0.0322,\,1.0,\,1.0,\,0.03,\,0.8) \end{aligned}$$

## G   LLM USAGE STATEMENT

LLMs were not involved in the writing or ideation of this work. Cursor was used as a coding IDE, and LLM tab completions were use to implement various functions, but the data and concepts were generated without LLM usage (except for the openwebtext dataset, which used a language embedder as part of the system in Appendix D.

