# OpenReview forum: "MuSED-FM: A Benchmark for Evaluating Multivariate Time Series Foundation Models"
_ICLR.cc/2026/Conference — ICLR 2026 Conference Withdrawn Submission_

### Official Review · Reviewer_RnsE · 2025-10-27

**Soundness:** 2
**Presentation:** 2
**Contribution:** 2
**Rating:** 2
**Confidence:** 4

**Summary:**

This paper introduces MUSED-FM, a benchmark suite and evaluation protocol for assessing multivariate time-series foundation models (TSFMs). It assembles a diverse collection of multivariate datasets and proposes two aggregate dataset-quality metrics — univariate predictability and multivariate predictability — intended to quantify a dataset’s usefulness for training or evaluating foundation models. The paper demonstrates the benchmark with empirical experiments on candidate TSFMs.

**Strengths:**

1.Timely and relevant: addresses the growing need to evaluate pre-trained / time-series foundation models.
2.The benchmark’s scale and dataset diversity are impressive.

**Weaknesses:**

1.The two aggregate metrics (univariate and multivariate predictability) are validated only via comparisons to “Random” and “Other” covariates; this is insufficient to demonstrate reliability. A more rigorous validation is required, which could include validating the contribution of individual sub-metrics.
2.The evaluation would be more insightful if it included comparisons against strong, non-foundation model baselines.
3.The paper lacks a direct comparison to existing multivariate time-series benchmarks. Further evidence is required to conclusively demonstrate the superiority or unique value of the proposed benchmark over prior work.

**Questions:**

see Weaknesses.

---

### Official Review · Reviewer_LubR · 2025-10-31

**Soundness:** 3
**Presentation:** 3
**Contribution:** 3
**Rating:** 6
**Confidence:** 4

**Summary:**

This work proposes a multi-domain benchmark to test whether the time-series foundation model (FM) actually uses cross-variable information. It uses a large and diverse multivariate corpus that was taken from four sources. In order to assess when multivariate context should help, the authors are proposing aggregated predictability metrics, which contain a univariate score that captures how predictable a target is from its own history and a multivariate score that estimates how much extra signal exists in covariates, built from classical tools and a simple DLinear forecaster. By using this benchmark, the authors are evaluating representative time series FMs and baselines and also showing a consistent and striking result. I found their contributions in two folds: (1) broad and careful constructions, and (2) an analysis framework that quantifies when multivariate help should exist, while revealing that SOTA models do not capitalize on it. I found the second point (2) underscores the need for better architectures and training strategies for true multivariate forecasting, which seems to be an important point in time-series forecasting in general.

**Strengths:**

As to the originality of the work, I found it a broad benchmark with a new and operational definition of multivariate help via aggregated predictability metrics. Beyond the real multivariate datasets, they're creatively adding combined correlated univariate collections and turning high-volume sequential sources (e.g., images, text) into multivariate time-series and introducing two synthetic generators, so from my understanding, there are settings where a cross-variable signal is guaranteed.
As to the quality, the dataset that was used is large and diverse, and the authors are clear on why each source class is included and what they're testing on. I also found the two synthetic families well-motivated, which shows the ground-truth multivariate structure for stress-testing models.
As to the clarity, I found the paper easy to follow and somehow easy-to-understanding, but I'd suggest providing a brief definition of your definition of foundation models in general. I mention it because, as you know, people have different definitions of a foundation model, and it could help other reviewers to understand your thoughts better if they're not working on this type of research.
As to the significance of the work, I think the core empirical result is interesting and important, because today's multivariate capable time series Foundation models usually fail to beat their own univariate variants, even in domains and settings where covariates provably carry signal. One notable comment that I'd like to raise is that the benchmark they're proposing quantifies " when multivariate should help," and it gives the community a concrete insight for progress and a room to test their ideas in this architecture. Given the scale, domain and diagnostic framing, I think this work is positioned well to become a standard for measuring real multivariate capability.

**Weaknesses:**

There are a couple of things that I noticed and would like to be more clarified.

From my understanding, the paper is mostly focusing on zero-shot evaluations, which can understate the model's ability to use cross-variable signal, and I personally think it'd be good to share a few-shot and/or even fine-tuning, as well as LoRA pass that uses the paper's own predictability scores to select the top-K covariates per target before training. I think if the multivariate still fails under light adaptation, the results could be stronger and more interesting for the community, and if it improves, the benchmark becomes a more useful guide for the design of the method and would be curious to hear the authors' thoughts on it

---
One more thing is the sequential to time-series construction, which uses tokenization, dimensionality reduction to a fixed set of variants and noising to match traditional time-series statistics. I personally think these steps may suppress cross-variable structure in ways that models can't exploit. I was wondering if the authors are considering sharing the exact transforms and adding stage-wise ablations? Maybe evaluating predictability and model performance before and after noising, and after each reduction step, as well as including negative controls such as phase randomization or within-domain variable permutation to confirm that gains aren't artifacts, could be a good approach.

---
My last concern and also comment/question is about data balance, which I think it'd deserves more clarification. From my understanding, the traditional slice is ~0.5B points across 37 datasets, while synthetic and combined contribute billions more, so I was wondering if the authors are considering sharing headline univariate vs multivariate results separately for each slice and if there is a possibility, weight by domain to avoid the synthetic share dominating the conclusion. Also, it seems to me that the paper intends to open-source code and metrics, so I think for a benchmark, this is crucial, so maybe releasing the preprocessing, splits, and relevant scripts could be useful. Please note that this is not a main concern, and I am not deducting any score because of this. It's a simple suggestion.

**Questions:**

I have two questions, and I also encourage the authors to check the weaknesses part as well

(1) I would be curious how well your predictability metrics track ground-truth multivariate signal on the synthetic generators, and could they miss nonlinear or conditional effects? I think sharing the correlation between the aggregated metric and the realized univariate to multivariate gain and a small set of modern measures (e.g., conditional transfer entropy) could help. Asking this because it would calibrate the metric and reduce the chance that datasets are flagged as multivariate-helpful when the signal is nonlinear or at least non-stationary.

---
(2) From my understanding, in your setup, you relate predictability to mean MAPE and your forecasting objective conditions only on past covariates and past target. I was wondering what the authors' thoughts if they rerun the univariate vs multivariate comparison with a metric suite (e.g., CRPS, MAE, RMSE) and a variant where known-future covariates are provided at prediction time? Asking this because if the univariate > multivariate result holds across metrics and with known-future inputs, it becomes much stronger.

---

### Official Review · Reviewer_hR9R · 2025-11-01

**Soundness:** 3
**Presentation:** 2
**Contribution:** 2
**Rating:** 4
**Confidence:** 3

**Summary:**

The paper introduces MuSED-FM, a large, curated benchmark and analysis suite designed to evaluate multivariate Time Series Foundation Models (TSFMs). MuSED-FM aggregates 45 datasets across 16 domains (claimed: ≈67 billion timesteps, ≈2.6M series, avg ~26 variates), supplements real-world collections with transformed sequential data and two kinds of synthetic multivariate data, and proposes aggregated “univariate” and “multivariate” predictability metrics

**Strengths:**

The dataset collection is impressively large and diverse (many domains, synthetic + transformed sequential data).
Introducing and combining multiple classical measures into an aggregate multivariate-predictability statistic is valuable: it helps disentangle dataset informativeness from model failure modes.

**Weaknesses:**

Large parts of MuSED-FM come from generated or transformed sources. While this is a strength, the manuscript lacks sufficiently detailed justification and validation that these transformed/synthetic series realistically mimic multivariate predictive structure found in real applications.

The manuscript builds a large benchmark by aggregating existing datasets and adding synthetic/converted sources, but it does not present any newly collected real-world dataset.

Figure 2’s layout places method names too close to the graphical elements, causing visual clutter.

The paper uses synthetic data and “transformed” sources (images→timeseries; SCM/dynamical system simulations) but does not adequately describe the generative procedures, parameter choices, noise models, or the range of settings tried.

**Questions:**

Addressing the three items above will materially strengthen the paper’s credibility and impact. In particular:

Either add new real data or convincingly demonstrate that the assembled+synthetic corpus truly captures the realistic phenomena the paper claims to study.

Fully describe the detailed synthetic data generation process.

---

### Official Review · Reviewer_bGZ9 · 2025-11-03

**Soundness:** 3
**Presentation:** 1
**Contribution:** 2
**Rating:** 2
**Confidence:** 5

**Summary:**

This work introduces MUSE-FM, a large-scale benchmark comprising 16 domains and 67 billion data points to evaluate Multivariate Time Series Foundation Models (TSFM). It further proposes new tools and metrics to assess multivariate predictability, revealing that current TSFM often fail to outperform univariate baselines despite multivariate correlations.

**Strengths:**

na

**Weaknesses:**

While the paper introduces MUSE-FM, a large-scale benchmark for evaluating Multivariate Time Series Foundation Models (TSFM), the writing lacks rigor and clear organization. The novelty is limited, as the work mainly presents a dataset and a set of evaluation metrics rather than proposing new methodological insights or theoretical contributions in multivariate time series modeling.

**Questions:**

Moreover, several key questions remain unanswered:

Covariate design: When extending from univariate to multivariate time series, how are covariates constructed or selected to ensure meaningful multivariate relationships? This is a critical issue, yet the paper does not provide sufficient clarification.

Heterogeneous data handling: How does the benchmark address differences in data types and scales, particularly when evaluating TSFM models that assume a single scale or homogeneous data structure?

---

### Note · Authors · 2025-11-20

I have read and agree with the venue's withdrawal policy on behalf of myself and my co-authors.